# An outer membrane porin-lipoprotein complex modulates elongasome movement to establish cell curvature in *Rhodospirillum rubrum*

Sebastian Pöhl [1,12], Giacomo Giacomelli[2,12], Fabian M. Meyer[2], Volker Kleeberg[3,10], Eli J. Cohen[4], Jacob Biboy[5], Julia Rosum[1], Timo Glatter [6], Waldemar Vollmer[5,7], Muriel C. F. van Teeseling[1,11], Johann Heider[1,8], Marc Bramkamp [2] & Martin Thanbichler [1,8,9] ✉

Curved cell shapes are widespread among bacteria and important for cellular motility, virulence and fitness. However, the underlying morphogenetic mechanisms are still incompletely understood. Here, we identify an outer-membrane protein complex that promotes cell curvature in the photo-synthetic species *Rhodospirillum rubrum*. We show that the *R. rubrum* porins Por39 and Por41 form a helical ribbon-like structure at the outer curve of the cell that recruits the peptidoglycan-binding lipoprotein PapS, with PapS inactivation, porin delocalization or disruption of the porin-PapS interface resulting in cell straightening. We further demonstrate that porin-PapS assemblies act as molecular cages that entrap the cell elongation machinery, thus biasing cell growth towards the outer curve. These findings reveal a mechanistically distinct morphogenetic module mediating bacterial cell shape. Moreover, they uncover an unprecedented role of outer-membrane protein patterning in the spatial control of intracellular processes, adding an important facet to the repertoire of regulatory mechanisms in bacterial cell biology.

Bacteria come in a variety of different shapes to ensure optimal fitness in the ecological niches they inhabit[1–3]. Their morphology is usually determined by a peptidoglycan cell wall, a complex mesh-like polymer composed of linear glycan strands that are crosslinked by short peptides[4,5]. The peptidoglycan layer surrounds the cytoplasmic membrane and forms a single large macromolecule, called the sacculus. This structure needs to be continuously remodeled to enable cell growth and division[1,6].

Peptidoglycan synthesis is achieved by a diverse set of synthetic and lytic enzymes that are typically combined into dynamic multi-protein complexes to ensure the close coordination of their activities[6,7]. Two widely conserved complexes involved in cell

[1]Department of Biology, University of Marburg, Marburg, Germany. [2]Institute of General Microbiology, Kiel University, Kiel, Germany. [3]Institut für Biologie II, University of Freiburg, Freiburg, Germany. [4]Department of Life Sciences, Imperial College London, London, UK. [5]Centre for Bacterial Cell Biology, Institute for Cell and Molecular Biosciences, Newcastle University, Newcastle upon Tyne, UK. [6]Mass Spectrometry and Proteomics Facility, Max Planck Institute for Terrestrial Microbiology, Marburg, Germany. [7]Institute for Molecular Bioscience, University of Queensland, Brisbane, QLD, Australia. [8]Center for Synthetic Microbiology (SYNMIKRO), Marburg, Germany. [9]Max Planck Fellow Group Bacterial Cell Biology, Max Planck Institute for Terrestrial Microbiology, Marburg, Germany. [10]Present address: Pädagogische Forschungsstelle Kassel, Kassel, Germany. [11]Present address: Institute of Microbiology, Friedrich-Schiller-Universität, Jena, Germany. [12]These authors contributed equally: Sebastian Pöhl, Giacomo Giacomelli. ✉e-mail: thanbichler@uni-marburg.de

morphogenesis are the elongasome and the divisome[6,8]. In bacteria that elongate by lateral growth, the elongasome is organized by the actin homolog MreB and moves around the circumference of the cell along helical paths, promoting cell elongation by dispersed incorporation of new peptidoglycan along the long axis of the cell[9]. The divisome, by contrast, assembles at the future division site, in a manner dependent on the tubulin homolog FtsZ, and mediates cell constriction as well as the formation of the two new cell poles during cytokinesis[10]. These two systems are largely sufficient to generate the prototypical straight rod-shaped morphologies observed for many commonly studied model species[6]. To establish more complex shapes, bacteria have evolved additional mechanisms that promote anisotropic cell wall growth, thereby for instance distorting the sacculus into curved structures or inducing the formation of cellular extensions such as stalks or buds[1].

In recent years, significant progress has been made in understanding the symmetry-breaking mechanisms that govern the establishment of complex morphological features. This is particularly true for the generation of curved and helical cell shapes, which are widespread among bacteria, promoting surface colonization[11], motility in viscous environments, and virulence[12–16]. The systems found to mediate curved morphology are highly diverse with respect to the proteins involved and their modes of action. However, as a common characteristic, they typically rely on polymer-forming proteins that associate with the cell envelope and then alter the mode or rate of peptidoglycan remodeling in their vicinity[17]. Prominent examples are the cytoplasmic coiled-coil-rich protein crescentin of *Caulobacter crescentus*[18,19] and the periplasmic CrvAB complex of *Vibrio cholerae*[20,21], which are both localized to the inner curve of the cell and thought to act by exerting a mechanical force on the cell envelope. The helical shape of *Helicobacter pylori*, by contrast, depends on the cytoplasmic bactofilin homolog CcmA, which localizes to the outer curve and interacts with several peptidoglycan hydrolases to locally relax the peptidoglycan meshwork[22–24]. Targeted cell wall hydrolysis at the outer curve is also critical for helical shape in *Bdellovibrio bacteriovorus*, but the spatial cues mediating the asymmetric localization of the enzyme involved are still unknown[25]. Finally, periplasmic flagella were found to be required for the dynamic helical morphology of *Borrelia burgdorferi*[26].

Curved or helical cell shapes are also common among members of the alphaproteobacterial family *Rhodospirillaceae*, a large group of organisms with significant ecological and biotechnological importance[27]. Their type species *Rhodospirillum rubrum* is a facultatively phototrophic bacterium that is widespread in aquatic environments[28–30]. It is commonly studied for its photosystem and its ability to fix nitrogen, grow on CO gas, produce hydrogen, and accumulate the bioplastic precursor polyhydroxybutyrate[27]. However, its cell biology is largely uninvestigated.

In this study, we report the identification of a conserved outer membrane porin-lipoprotein complex that governs the establishment of cell curvature in *R. rubrum* cells. We show that the lipoprotein PapS[31] forms stable helical assemblies at the outer curve of the cell that bridge the outer membrane and the peptidoglycan layer, with defects in its bridging activity leading to cell straightening. We further demonstrate that the helical localization pattern of PapS is mediated by the two porins Por41 and Por39 of *R. rubrum*[31], which establish helical arrays in the outer membrane that colocalize with PapS. Importantly, mutations that disrupt porin localization or the formation of porin-PapS complexes lead to the delocalization of PapS and the loss of cell curvature, indicating a direct link between the helical arrangement of porin-PapS complexes and the establishment of curved cell morphology. Analyzing elongasome motion, we then demonstrate that PapS assemblies act as roadblocks that bias the distribution of elongasome complexes to the outer cell curve, thus generating a longitudinal zone of elevated peptidoglycan biosynthesis that distorts the cell wall cylinder into a

helical shape. Together, these findings identify a previously unknown morphogenetic module that controls the spatiotemporal dynamics of the basic cell elongation machinery to generate curved cell shape. Moreover, they reveal a striking example of outer-membrane protein patterning and highlight a key role of this process in the regulation of cell wall biosynthetic complexes in the inner layers of the cell envelope.

## Results

### The conserved lipoprotein PapS is required for cell curvature in *R. rubrum*

Previous work has shown that the two outer-membrane porins Por39 and Por41 of *R. rubrum* form a highly stable complex with a so-far uncharacterized protein (PAP) containing a putative OmpA-like peptidoglycan-binding domain[31,32], now renamed PapS (*P*orin-*a*ssociated *p*rotein mediating *s*piral cell shape). PapS (*Rru_A3328* in type strain S1[33]) is one of the most abundant proteins within the cell, ranking at third position behind the peptidoglycan-associated lipoprotein Pal[34] and porin Por41 (Fig. 1a and Supplementary Data 1). Its N-terminal region features a putative Sec signal peptide with a predicted signal peptidase II cleavage site and no inner-membrane retention signal[35–37], suggesting that it is an outer-membrane lipoprotein. To verify the periplasmic localization of PapS, we performed a β-lactamase (Bla) assay, using the fact that Bla fusion proteins only confer resistance to the β-lactam ampicillin if translocated to the periplasm. Cells producing a PapS-Bla fusion showed high-level ampicillin resistance, confirming its targeting to the periplasmic space (Fig. 1b and Supplementary Fig. 1a, b). Protein structure prediction with AlphaFold2[38] and protein domain analysis indicate that the processed form of PapS comprises an N-terminal domain of unknown function and a C-terminal OmpA-like peptidoglycan-binding domain[39,40], connected by a short non-structured linker (Fig. 1c and Supplementary Fig. 1c). OmpA-like domains specifically interact with the meso-diaminopimelic acid (mDAP) residue in the peptide side chain of uncrosslinked peptidoglycan subunits[39,40]. We verified that the isolated OmpA-like domain of PapS bound mDAP in vitro, albeit with lower affinity than the previously characterized OmpA homolog of *Acetobacter baumannii*[39,40] (Fig. 1d). Its association with the outer membrane, the Por39/41 complex, and the peptidoglycan layer thus links PapS tightly to the cell envelope, consistent with the previous observation that it is recovered in the detergent-insoluble cell wall fraction of *R. rubrum* lysates[31].

To clarify the function of PapS, we generated an in-frame deletion in its gene. Surprisingly, ΔpapS cells no longer showed the typical curved morphology of *R. rubrum* cells but instead had a straight, rod-like shape (Fig. 1e and Supplementary Fig. 1d), as also supported by a quantification of wild-type and mutant cell sinuosities (Fig. 1f). In addition, while leaving the growth rate unaffected, the loss of PapS led to moderate decrease in the cell yield, indicating a reduction in cellular fitness (Supplementary Fig. 1e). The morphological defect of the ΔpapS mutant was fully reversed by ectopic expression of *papS* under the control of its native promoter from a low-copy-number plasmid (Fig. 1f and Supplementary Fig. 1a, d). However, the plasmid-bearing mutant still showed lower cell yields (Supplementary Fig. 1e,f), possibly because it accumulated PapS to lower levels than wild-type cells. These results demonstrate that PapS is critical for cell curvature and proper growth in *R. rubrum*.

To further illuminate the role of PapS, we purified peptidoglycan sacculi from wild-type and ΔpapS cells and analyzed their morphology. Wild-type sacculi retained a curved shape, indicating that cell curvature in *R. rubrum* is established by asymmetric peptidoglycan remodeling rather than transient mechanical bending of the cell envelope (Fig. 1g, h). The sinuosity of these sacculi was lower than that of living cells, which may be explained by anisotropic shrinking in the absence of the turgor pressure, caused by the distinct orientation and elasticity of glycan chains and peptide bridges within the peptidoglycan

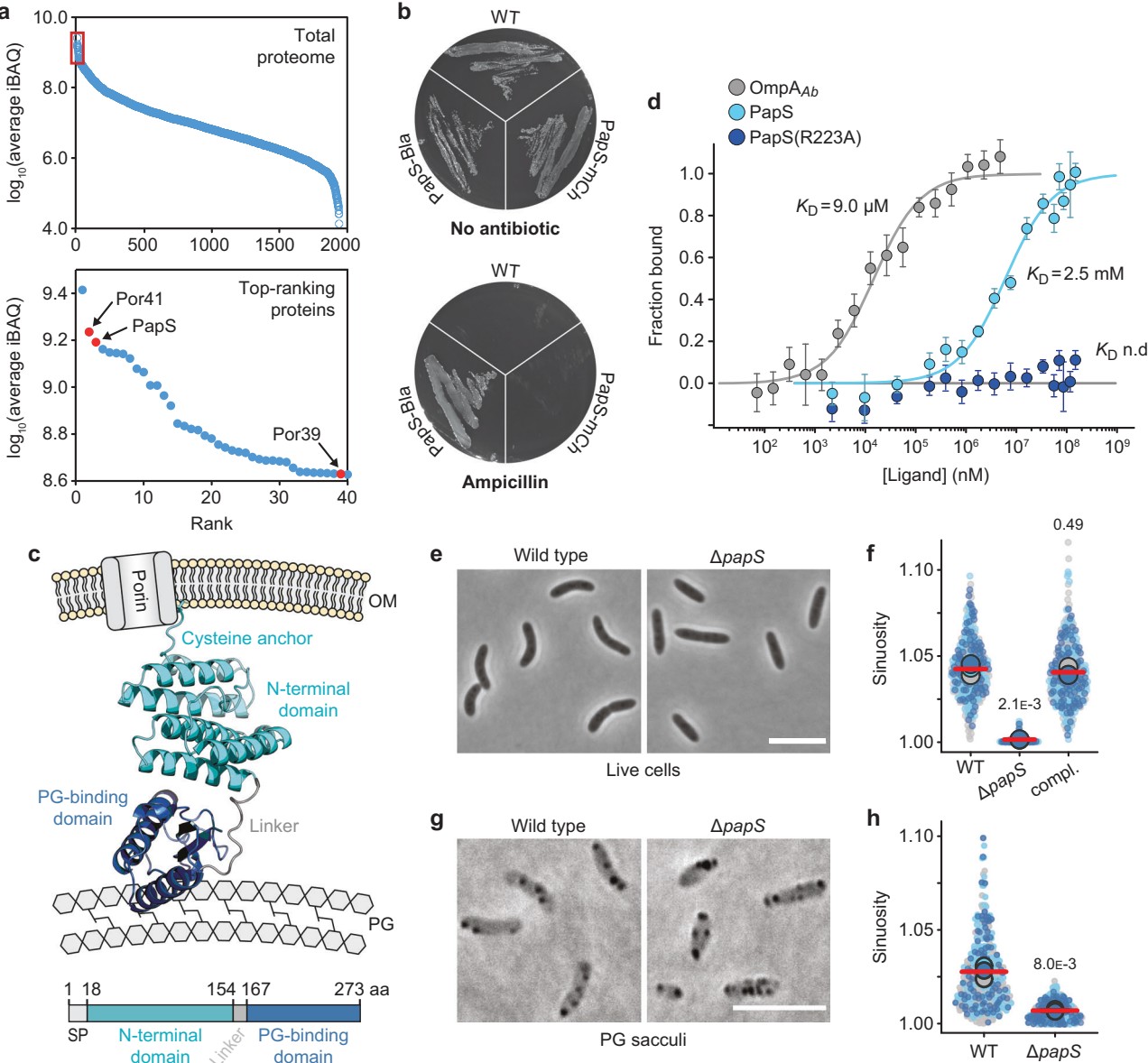

**Fig. 1 | PapS has a decisive role in _R. rubrum_ cell curvature. a** Total protein abundance in _R. rubrum_ wild-type (S1) cells, as estimated on the basis of iBAQ (intensity-based absolute quantification) values derived from whole-cell proteome analysis. The graph on the left shows all 1943 proteins detected ranked according to their cellular abundance level. The red box indicates the 40 most abundant proteins, whose levels are further detailed in the graph on the right. Relevant proteins are highlighted in red. **b** β-lactam resistance assay confirming the periplasmic localization of PapS. Cells of strains S1 (WT), SP231 (_papS-bla_) and SPO1 (_papS-mCherry_) were streaked on solid medium without or with ampicillin (200 μg ml⁻¹). Ampicillin resistance indicates the translocation of PapS-Bla to the periplasm. **c** Structural model of _R. rubrum_ PapS, generated with AlphaFold2[38] and shown in the predicted cell envelope context (OM: outer membrane, PG: peptidoglycan). The N-terminal domain is shown in turquoise, the linker in grey and the OmpA-like peptidoglycan-binding domain in dark blue. The scheme at the bottom shows the position of the different domains in the amino acid sequence of PapS (SP: signal peptide, aa: amino acids). **d** Microscale thermophoresis analysis investigating the interaction of the OmpA-like peptidoglycan-binding domains (100 nM) of _R. rubrum_ PapS and PapS_{R223A} (amino acids 156-273) and _A. baumannii_ OmpA (amino acids 221-339) with increasing concentrations of mDAP. Data represent the mean

(±SD) of 3 independent experiments. The resulting $K_D$ values are indicated next to the respective binding curves. **e** Morphology of _R. rubrum_ wild-type (S1) and Δ_papS_ (JR52) cells, visualized by phase-contrast microscopy. Bar: 5 μm. **f** Superplots showing the distribution of cell sinuosities in populations of _R. rubrum_ wild-type (S1) and Δ_papS_ (JR52) cells and of Δ_papS_ cells expressing a complementing copy of _papS_ from a low-copy number plasmid (compl.; JR54). Small dots represent the data obtained in three independent biological replicates (shown in dark blue, light blue and grey; n = 100 cells per replicate). Large dots represent the mean value of the three datasets. The red horizontal line indicates the average of these three mean values. The statistical significance (_p_ value) of differences between the wild type and the two mutant strains is indicated (unpaired two-sided Welch's _t_-test). **g** Purified peptidoglycan sacculi of _R. rubrum_ wild-type (S1) and Δ_papS_ (JR52) cells, visualized by phase-contrast microscopy. Black dots likely represent poly-hydroxybutyrate granules[92]. Bar: 5 μm. **h** Superplots showing the sinuosities of peptidoglycan sacculi isolated from _R. rubrum_ wild-type (S1) and Δ_papS_ (JR52) cells. The data are presented as described in panel (**f**) (n = 100 cells per replicate). The statistical significance (_p_ value) of differences between the wild type and the two mutant strains is indicated (unpaired two-sided Welch's _t_-test). Source data are provided as a **Source Data** file.

meshwork[41]. Sacculi from Δ*papS* cells, by contrast, were again straight, supporting a role of PapS in the spatiotemporal control of peptidoglycan biosynthesis (Fig. 1g, h).

After having identified PapS as a key morphogenetic factor in *R. rubrum*, we performed database searches to analyze its phylogenetic distribution. Clear PapS homologs sharing both the PapS-specific N-terminal domain and the OmpA-like C-terminal domain turned out to be widespread in several lineages of the Alphaproteobacteria, in particular the order *Rhodospirillales*, with many species containing more than one PapS protein (Supplementary Fig. 2 and Supplementary Data 2). Given the high abundance of curved and helical species in the *Rhodospirillales*, these results suggest that the curvature-inducing activity of PapS is not restricted to *R. rubrum* but conserved in a wide variety of bacteria.

## PapS forms a static, cell envelope-associated helical structure in the periplasm

To clarify the mode of action of PapS, we first determined its localization pattern. To this end, we generated an *R. rubrum* strain in which the native *papS* gene was replaced with an allele encoding a PapS variant fused to the monomeric green fluorescent protein mNeon-Green (mNG)[42]. The fusion protein was functional and still mediated cell curvature, although the sinuosity of the cells was slightly reduced, potentially due to steric hindrance by the fluorescent tag or the presence of degradation products (Supplementary Fig. 3a, b). When visualized by epifluorescence microscopy, PapS-mNG was exclusively detected at the outer curve and the poles of the cell, i.e. regions of positive Gaussian curvature (Fig. 2a). A significant enrichment of the fusion protein in these regions was confirmed by a quantitative comparison of the integrated signal intensities at the inner and outer curve (Fig. 2b). To resolve the pattern formed in more detail, we treated cells with the cell division inhibitor cephalexin[43], thereby inducing the formation of helical cell filaments that still showed wild-type cell curvature (Supplementary Fig. 3c, d). Subsequent super-resolution imaging with three-dimensional structured illumination microscopy (3D-SIM) revealed that PapS-mNG formed a continuous ribbon-like structure that specifically lined the outer curve of the cells (Fig. 2c and Supplementary Movie 1). The helical arrangement of PapS and its critical role in the establishment of cell curvature suggested a direct involvement of this protein in *R. rubrum* cell shape determination.

To investigate the localization dynamics of PapS, we next monitored cells producing the PapS-mNG fusion over the course of the cell cycle (Fig. 2d and Supplementary Movie 2). New-born cells typically showed a strong signal at the old cell pole, which gradually decreased towards the new pole. As the cells elongated, the PapS structure increased in length accordingly, with the new-pole signal becoming more intense and additional, fainter fluorescence appearing in the midcell region. These results suggest that new PapS-mNG molecules are incorporated along the entire length of the structure and stay fixed in place after their insertion, thus accumulating over time until a certain saturation level is reached. The static nature of the PapS structure is supported by fluorescence-recovery-after-photobleaching (FRAP) experiments, which showed that the vast majority of PapS-mNG molecules were immobile, with essentially no recovery of the signal within a 15-min time frame after bleaching (Fig. 2e and Supplementary Movie 3).

Our findings indicated that PapS formed a stable, ribbon-like assembly in the cell envelope. To clarify the architecture of this structure, we generated a strain in which PapS was fused to the photoactivatable fluorescent protein PAmCherry2 (PAmCh)[44] (Supplementary Fig. 3a, b) and then investigated the subcellular distribution of the fusion protein using single-molecule localization microscopy (SMLM). Consistent with the proteomics data (Fig. 1a), we found that PapS-PAmCh was highly abundant (>15,000 copies per cell) and densely packed in the outer curve, with the majority of molecules

placed within a distance of 20 nm from each other (Fig. 2f, g). This result raised the possibility that PapS was able to self-interact and thus assemble into helical polymers. However, purified PapS was exclusively monomeric when analyzed by gel filtration in vitro (Supplementary Fig. 4). Moreover, when we monitored the localization pattern of PapS-mNG in cells that were gently lysed on an agarose pad[20], the ribbon structures formed disintegrated immediately upon rupture of the cell envelope, with the fusion protein remaining attached to the envelope fragments (Fig. 2h and Supplementary Movie 4). Together, these observations argue against the existence of stable PapS polymers and confirm a close association of PapS with the outer membrane and the peptidoglycan layer.

## Peptidoglycan binding is critical for PapS function

PapS is predicted to interact with the cell envelope through both its outer-membrane lipid anchor and its OmpA-like peptidoglycan-binding domain (Fig. 1c). To determine the importance of these features for PapS function, we generated a PapS variant in which the native lipoprotein signal sequence was replaced with the signal sequence of the soluble periplasmic protein DipM from *Caulobacter crescentus*[45–47] (Fig. 3a). In addition, we constructed a variant in which we exchanged a conserved arginine residue (R223) in the OmpA-like domain whose equivalent in *A. baumannii* OmpA was shown to be critically involved in mDAP and, thus, cell wall binding[40] (Fig. 3b). In vitro analysis verified that the mutant domain no longer showed mDAP-binding activity (Fig. 1d). When fused to mNeonGreen and produced in trans in the Δ*papS* background (Supplementary Fig. 5a), the protein defective in outer-membrane binding still induced appreciable cell curvature (Fig. 3c). The variant carrying the mutant OmpA-like domain, by contrast, barely showed any cell-bending activity (Fig. 3c). These results indicate that peptidoglycan binding is required for PapS to mediate cell curvature, whereas outer-membrane binding only has an auxiliary role in its function.

Importantly, despite their different functionalities, both mutant proteins still displayed the characteristic helical localization pattern when analyzed by fluorescence microscopy (Fig. 3d and Supplementary Movies 5, 6, 7), with a clear, albeit reduced enrichment of the signal in the outer curve of the cells (Fig. 3e). However, single-particle tracking studies revealed that the corresponding PAmCh fusions (Supplementary Fig. 5b, c) showed marked differences in their diffusional properties. Surprisingly, the loss of the lipoprotein signal sequence led to a moderate decrease in protein mobility, suggesting that membrane attachment promotes the movement of PapS molecules within the periplasm. The mutation in the OmpA-like domain, by contrast, markedly increased the proportion of mobile molecules (Fig. 3f and Supplementary Data 3). The static behavior of PapS structures thus appears to be explained, at least in part, by its association with the peptidoglycan layer. However, neither outer-membrane nor peptidoglycan binding is required to establish the helical localization pattern of PapS, pointing to the existence of a, to that point, unknown localization factor.

## Por39 forms a helical structure that colocalizes with PapS

Porins Por39 (Rru_A2212) and Por41 (Rru_A2211) of *R. rubrum* (Fig. 4a) had previously been shown to form stable complexes with PapS in vitro[31], opening the possibility that they were involved in the mechanism of PapS-dependent cell curvature. To test this hypothesis, we set out to determine the localization patterns of the two porins by replacing their native genes with alleles encoding mCherry sandwich fusions (Fig. 4b). While this approach was unsuccessful for *por41*, we readily obtained a strain expressing an *mCh-por39* fusion (Supplementary Fig. 6a, b). When visualized by fluorescence microscopy, the fusion protein formed helical filamentous assemblies reminiscent of those observed for PapS (Fig. 4c), with a significant enrichment of the signal in the outer curve of the cell (Fig. 4d). SMLM analysis of a strain

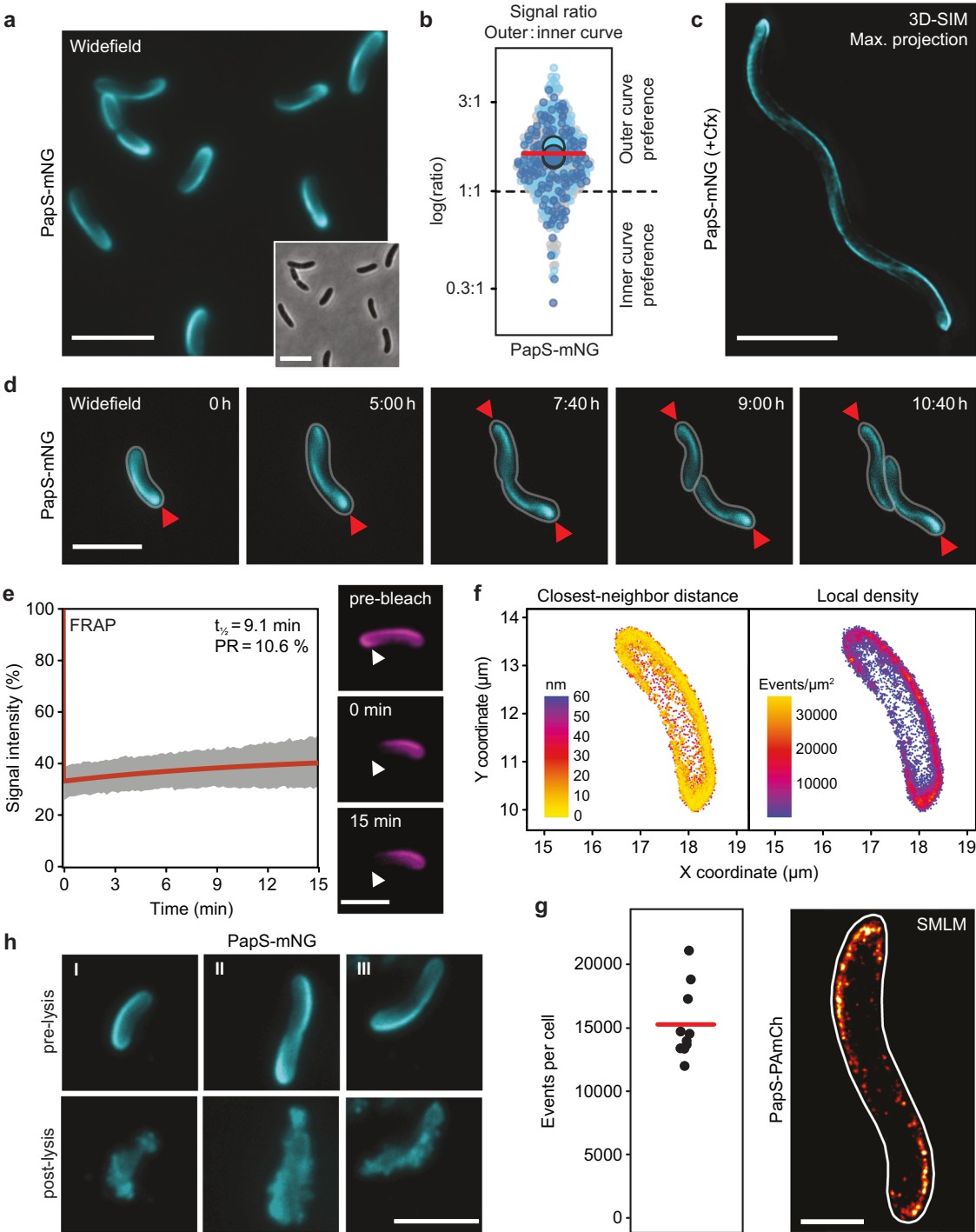

**Fig. 2 | PapS localizes to the outer curve of the cell. a** Localization of PapS-mNG (SP02) by widefield epifluorescence microscopy. Bar: 5 μm. **b** Superplots showing the outer-versus-inner-curve ratios of the PapS-mNG signal in the cell populations analyzed in (**a**). The data are presented as described for Fig. 1f. **c** Localization of PapS-mNG in *R. rubrum* cells (SP02) elongated by cefalexin (Cfx) treatment, as determined by three-dimensional structured illumination microscopy (3D-SIM). Bar: 5 μm. **d** Time-lapse series showing the formation of PapS ribbons in *R. rubrum* cells producing PapS-mNG (SP02). Arrowheads indicate the position of drop-like PapS accumulations at the cell poles. Bars: 5 μm. **e** FRAP analysis of PapS-mCh produced in place of the wild-type PapS protein (JR56). A part of the PapS structure was bleached, and the recovery of the signal was followed over time. The graph shows the average relative integrated signal intensity of the bleached region normalized to the average total signal intensity in the cell as a function of time (*n* = 5 cells; t₁/₂: recovery half-time, PR: percentage recovery). The fluorescence images on the right show a representative cell imaged directly before (pre-

bleach) and right after (*t* = 0 min) photobleaching and at the end of the experiment (*t* = 15 min). The arrowhead indicates the bleached region. Bar: 5 μm. **f** Single-molecule localization microscopy (SMLM) analysis of *R. rubrum* producing PapS-PAmCh in place of the native PapS protein (SP04). The heat maps indicate the closest-neighbor distance and the local density of PapS-PAmCh molecules. Shown are the results of a representative cell (*n* = 10 cells). **g** Pseudo-fluorescence image of a representative *R. rubrum* cell producing PapS-PAmCh (SP04), obtained by SMLM analysis. The graph gives the number of individual fluorescent particles (events) detected per cell (*n* = 10 cells). Bar: 1 μm.
**h** Disruption of PapS ribbons upon gentle cell lysis. Exponentially growing *R. rubrum* cells producing PapS-mNG (SP02) were transferred onto nutrient pads containing lytic agents. Shown are images of three representative cells (I-III) taken right before and after lysis. Bar: 3 μm. The experiment was performed four times independently with similar results. Source data are provided as a **Source Data** file.

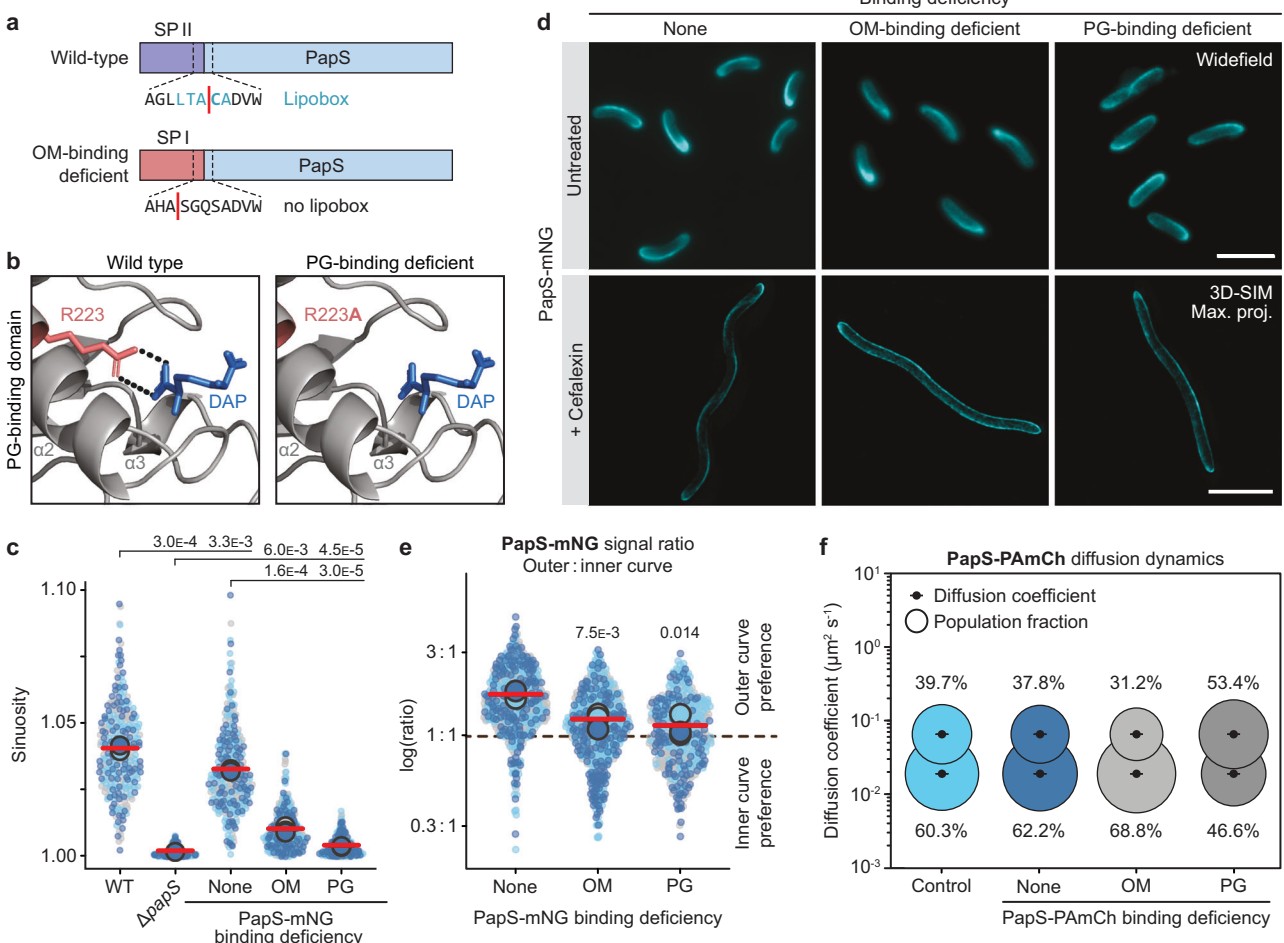

**Fig. 3 | Outer membrane and peptidoglycan binding are critical for PapS function. a** Scheme showing the wild-type PapS protein with its native PapS lipobox (top) and a chimera in which the lipoprotein signal sequence (SP II) is replaced by the signal sequence of the soluble periplasmic protein DipM from *C. crescentus* (SP I). The signal peptidase cleavage sites are indicated by a red line. **b** Predicted diaminopimelate (DAP)-binding sites of the OmpA-like domain of *R. rubrum* PapS (left) and a mutant (R223A) variant lacking an arginine residue implicated in peptidoglycan binding (right). The structures were generated by aligning AlphaFold2[38] models of the respective OmpA-like domains with the crystal structure of the peptidoglycan-binding domain of *Acinetobacter baumannii* OmpA in complex with DAP[40]. Residue R223 is depicted in red and DAP is shown in blue. Dashed lines indicate a bidentate salt bridge. **c** Superplots showing the distribution of cell sinuosities in populations of *R. rubrum* cells producing wild-type PapS-mNG (SP136), a PapS-mNG variant lacking the outer-membrane lipid anchor (SP137) and a PapS-mNG variant (R223A) defective in peptidoglycan binding (SP145) from a low-copy plasmid. Wild-type (S1) and Δ*papS* (JR52) cells were analyzed as controls. The data are presented as described for Fig. 1f (*n* = 100 cells per replicate). The statistical significance (*p* value) of differences between strains is indicated (unpaired two-sided Welch's *t*-test). The data represent the results of three biological replicates. **d** Localization of different PapS-mNG variants in the Δ*papS* background. Cells of a Δ*papS* mutant producing wild-type PapS-mNG (SP136), a PapS-mNG variant lacking the outer-membrane lipid anchor (OM; SP137) or a PapS-mNG variant (R223A)

defective in peptidoglycan binding (PG; SP145) from a low-copy plasmid were imaged after cultivation in the absence (untreated) or presence (+) of the cell division inhibitor cefalexin. Untreated cells were imaged by widefield epifluorescence microscopy, cefalexin-treated cells by 3D-SIM. Shown are representative cells (*n* = 3 biological replicates). Bar: 5 μm. **e** Superplots showing a quantification of the outer-versus-inner-curve PapS-mNG signal ratio in the cells described in panel (**d**). The data are presented as described for Fig. 1f. The statistical significance of differences between the wild type and the two mutant strains is indicated (unpaired two-sided Welch's *t*-test). Data represent the results of three biological replicates. **f** Bubble plots showing the apparent single-particle diffusion rates of the indicated PapS-PAmCh variants. Cells carrying *papS-mNG* in place of the endogenous *papS* gene (control; SP04) or Δ*papS* cells producing wild-type PapS-PAmCh (SP165), a PapS-PAmCh variant lacking the outer-membrane lipid anchor (SP166) or a PapS-PAmCh variant (R223A) defective in peptidoglycan binding (SP167) from a low-copy plasmid were subjected to single-particle tracking (SPT) analysis. The diffusive and immobile populations have mean apparent diffusion coefficients of 0.0657 μm² s⁻¹ and 0.0191 μm² s⁻¹, respectively. Note that for all single-particle tracking analyses presented in this study, the apparent diffusion coefficients of the two populations were determined globally and then constrained to be equal in the different conditions to enable a more direct comparison of the relative population sizes. The number of cells analyzed per strain is given in Supplementary Data 3. Source data are provided as a **Source Data** file.

expressing *PAmCh-por39* showed that the molecules in these structures were also densely packed and, in most part, located at a distance of less than 20 nm from each other (Fig. 4e). Their diffusional behavior was similar to that of the PapS-PAmCh fusion and characterized by a large (63%) proportion of largely immobile protein (Fig. 4f and Supplementary Data 3). Colocalization studies in a strain producing both mCh-Por39 and PapS-mNG (Supplementary Fig. 6a–c) confirmed that the signals of the two fusion proteins were essentially superimposable,

strongly suggesting a role of porins in PapS-mediated cell morphogenesis (Fig. 4g, Supplementary Movie 8 and Supplementary Data 4).

The colocalization of PapS and Por39 suggested that PapS could act as a localization factor for the porin complex. To test this possibility, we re-analyzed the localization pattern of mCh-Por39 in the Δ*papS* background (Supplementary Fig. 6a, b). Interestingly, the fusion protein retained its helical arrangement in the straight mutant cells, indicating that the porin complex is positioned independently of PapS

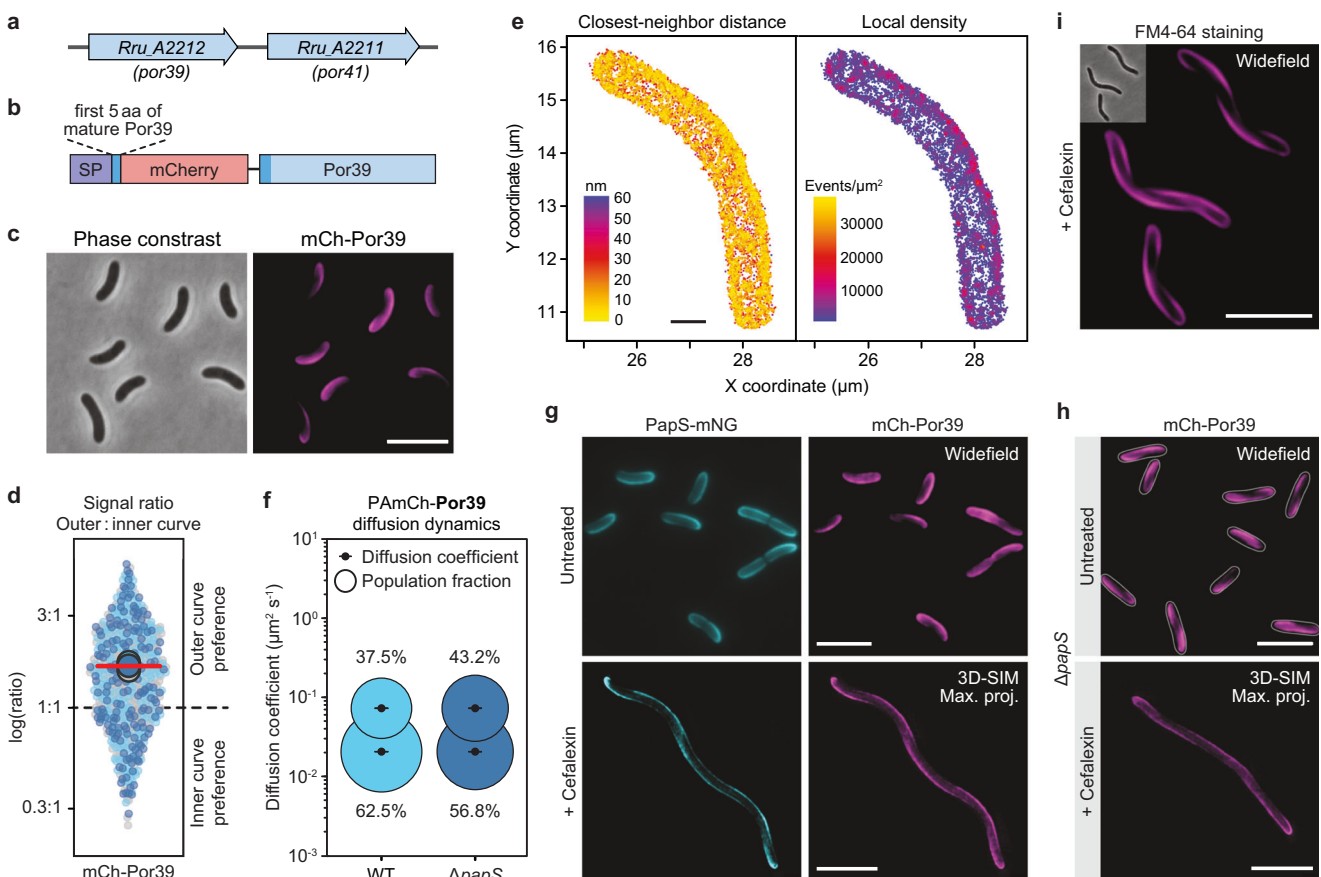

**Fig. 4 | Por39 colocalizes with PapS and shows a PapS-independent helical localization pattern. a** Chromosomal organization of the *por39* and *por41* genes. **b** Scheme showing the domain organization of the mCh-Por39 sandwich fusion protein, comprising the signal sequence (SP) and the following five amino acids of the unprocessed Por39 protein fused to the fluorescent protein mCherry and the processed (periplasmic) form of Por39. **c** Localization pattern of mCh-Por39. Cells producing a mCh-Por39 sandwich fusion protein in place of the native Por39 protein (SP222) were imaged by phase contrast and widefield epifluorescence microscopy. Shown are representative cells (*n* = 3 biological replicates). Bar: 5 μm. **d** Outer-versus-inner-curve mCh-Por39 signal ratio in populations of the cells described in **c**. The data are presented as described for Fig. 1f (*n* = 3 biological replicates). **e** SMLM analysis of *R. rubrum* producing PAmCh-Por39 in place of the native PapS protein (SP173). The heat maps indicate the closest-neighbor distance and the local density of PAmCh-Por39 molecules. Shown are the results of a representative cell (*n* = 20 cells). **f** Bubble plots showing the apparent single-particle diffusion rates of PAmCh-Por39 in the wild-type (SP173) and Δ*papS* (SP177)

backgrounds. The diffusive and immobile populations have mean apparent diffusion coefficients of 0.0625 μm² s⁻¹ and 0.0174 μm² s⁻¹, respectively. The number of cells analyzed is given in Supplementary Data 3. **g** Colocalization of PapS-mNG and mCh-Por39 in cells (SP06) grown in the absence (untreated) or presence (+) of the cell division inhibitor cefalexin. Untreated cells were imaged by widefield epifluorescence microscopy, cefalexin-treated cells by 3D-SIM. Shown are representative cells (*n* = 3 biological replicates). Bar: 5 μm. The area overlap of the PapS-mNG and mCh-Por39 signals in the 3D-SIM images is 90.3% (see also Supplementary Data 4). **h** Localization of mCh-Por39 in the Δ*papS* background. Cells (SP12) were cultivated in the absence (untreated) or presence (+) of cefalexin prior to imaging. Shown are representative cells analyzed by widefield epifluorescence microscopy (untreated) and 3D-SIM (+ Cefalexin) (*n* = 3 biological replicates). Bar: 5 μm. **i** FM4-64 staining of *R. rubrum* wild-type (S1) cells. Shown are representative cells (*n* = 3 biological replicates), analyzed by widefield microscopy. Bar: 5 μm. Source data are provided as a **Source Data** file.

(Fig. 4h and Supplementary Movie 9). However, in the absence of their interaction partner, porin molecules showed an increased mobility at the single-particle level, pointing to an inhibitory role of the peptidoglycan-associated PapS protein on porin diffusion (Fig. 4f). Interestingly, in both curved wild-type cells and straight Δ*papS* cells, the outer-membrane stain FM4-64 formed a helical pattern similar to that of the porin(-PapS) complexes (Fig. 4i and Supplementary Fig. 6d). Together, these results suggest an intrinsic helical organization of outer-membrane proteins and lipids in *R. rubrum*.

**The spatial organization of porins determines PapS localization**
Next, we aimed to clarify whether the helical arrangement of porin complexes was the driving force behind the formation of helical PapS structures. Since it was not possible to delete *por39* and *por41*, indicating an essential role of the two porins in *R. rubrum*, we aimed to identify and substitute amino acid residues that were specifically involved in their localization but not required for their function in

the exchange of ion and small-molecules across the outer membrane. A primary sequence alignment of all Por39 and Por41 homologs available in the databases revealed a highly conserved motif in the first periplasmic loop of the two proteins (Fig. 5a), which is predicted to reside close to the inter-subunit interface in the porin complex (Supplementary Fig. 7b, c). To test the functional relevance of this motif, we replaced the native porin genes with alleles encoding porin variants in which the aspartic acid residue (D71) in the center of the loop was replaced by serine (Supplementary Fig. 8a). When introduced into Por39, the substitution did not have any noticeable effect on cell curvature (Fig. 5b, c). Moreover, an mCh-Por39_{D71S} fusion still formed helical structures, which colocalized with PapS-mNG assemblies in the outer curve of the cell (Supplementary Fig. 8b–e, Supplementary Movie 10 and Supplementary Data 4). Cells carrying the mutant *por41_{D71S}* allele, by contrast, had completely lost their curved morphology (Fig. 5b, c). In this background, mCh-Por39 was no longer enriched in the outer curve but evenly dispersed within the

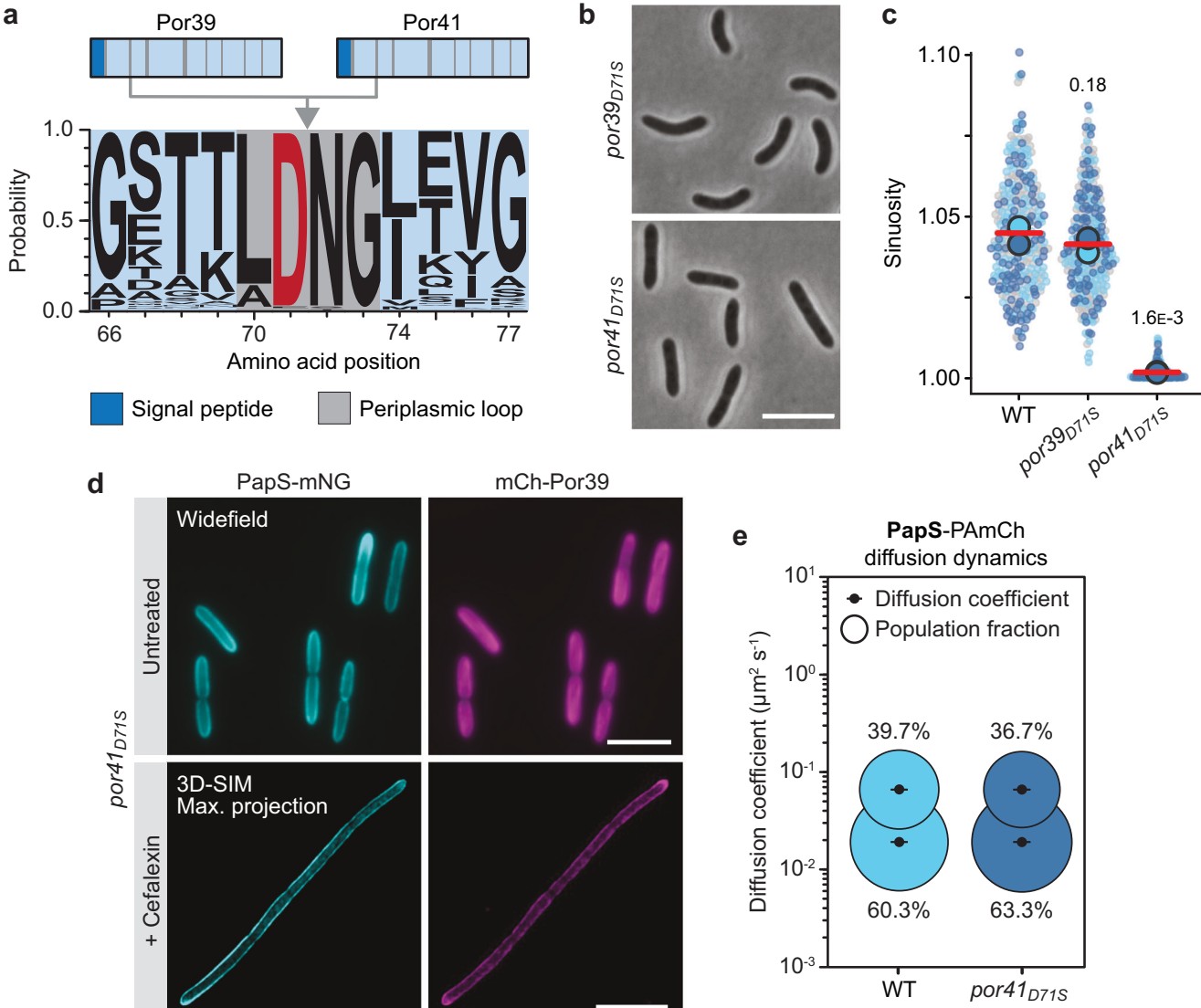

**Fig. 5 | Porin localization determines the formation of helical PapS ribbons.**
**a** Scheme showing the locations of the periplasmic loops (vertical grey lines) in the Por39 and Por41 polypeptides and the amino acid conservation of periplasmic loop 2, determined by aligning the sequences of all Por39/41 homologs identified in the nr database (NCBI) by BLAST analysis. The aspartate residue mutated in this study is highlighted in red. **b** Morphology of *R. rubrum* cells carrying the $por39_{D71S}$ (SP215) or $por41_{D71S}$ (SP150) alleles in place of the respective wild-type genes, imaged by phase-contrast microscopy. Bar: 5 μm. **c** Superplots showing the distribution of cell sinuosities in populations of *R. rubrum* wild-type (S1), $por39_{D71S}$ (SP215) and $por41_{D71S}$ (SP150) cells. The data are presented as described for Fig. 1f ($n = 100$ cells per replicate). The statistical significance (*p* value) of differences between the wild type and the two mutant strains is indicated (unpaired two-sided Welch's *t*-test).

**d** Colocalization of PapS-mNG and mCh-Por39 in the $por41_{D71S}$ background. Cells (SP131) were cultivated in the absence (untreated) or presence (+) of cefalexin prior to imaging. The images show representative cells analyzed by widefield epifluorescence microscopy (untreated) and 3D-SIM (+ Cefalexin) ($n = 3$ biological replicates). Bar: 5 μm. The area overlap of the PapS-mNG and mCh-Por39 signals in the 3D-SIM images is 82.5% (see also Supplementary Data 4). **e** Bubble plot showing the apparent single-molecule diffusion rates of PapS-PAmCh in the wild-type (SP04) and $por41_{D71S}$ (SP183) backgrounds. The data for PapS-PAmCh in the wild-type background are replotted from Fig. 3f as a reference. The diffusive and immobile populations have mean apparent diffusion coefficients of 0.0657 μm² s⁻¹ and 0.0191 μm² s⁻¹, respectively. The number of cells analyzed is given in Supplementary Data 3. Source data are provided as a **Source Data** file.

cell envelope, suggesting that changes in the conserved periplasmic loop of Por41 disrupt the helical arrangement of porins in *R. rubrum* (Fig. 5d, Supplementary Fig. 8c–e). Importantly, colocalization studies showed that this change in the porin pattern was accompanied by a loss of PapS-NG ribbons (Fig. 5d, Supplementary Fig. 8c–e, Supplementary Movie 11 and Supplementary Data 4). However, the diffusion behavior of PapS-PAmCh was unaffected in the $por41_{D71S}$ mutant (Fig. 5e, Supplementary Fig. 8c, e and Supplementary Data 3), suggesting that the D71S exchange did not disrupt the porin-PapS complex (see also below). Notably, the loss of porin localization did not affect the helical staining pattern of FM4-64, suggesting that the outer membrane of *R. rubrum* has an intrinsic, porin-independent

helical organization (Supplementary Fig. 6d). Together, these results demonstrate that the proper spatial arrangement of porins is critical for the formation of helical PapS assemblies, which in turn is required for the establishment of curved cell shape. In addition, they reveal a dominant role of Por41 over Por39 in the PapS-dependent morphogenetic system, which correlates with its considerably higher abundance in the cell.

## Porin binding drives the establishment of helical PapS structures

To study the dependence of PapS localization on the two porins in a more direct manner, we aimed to determine the porin-PapS

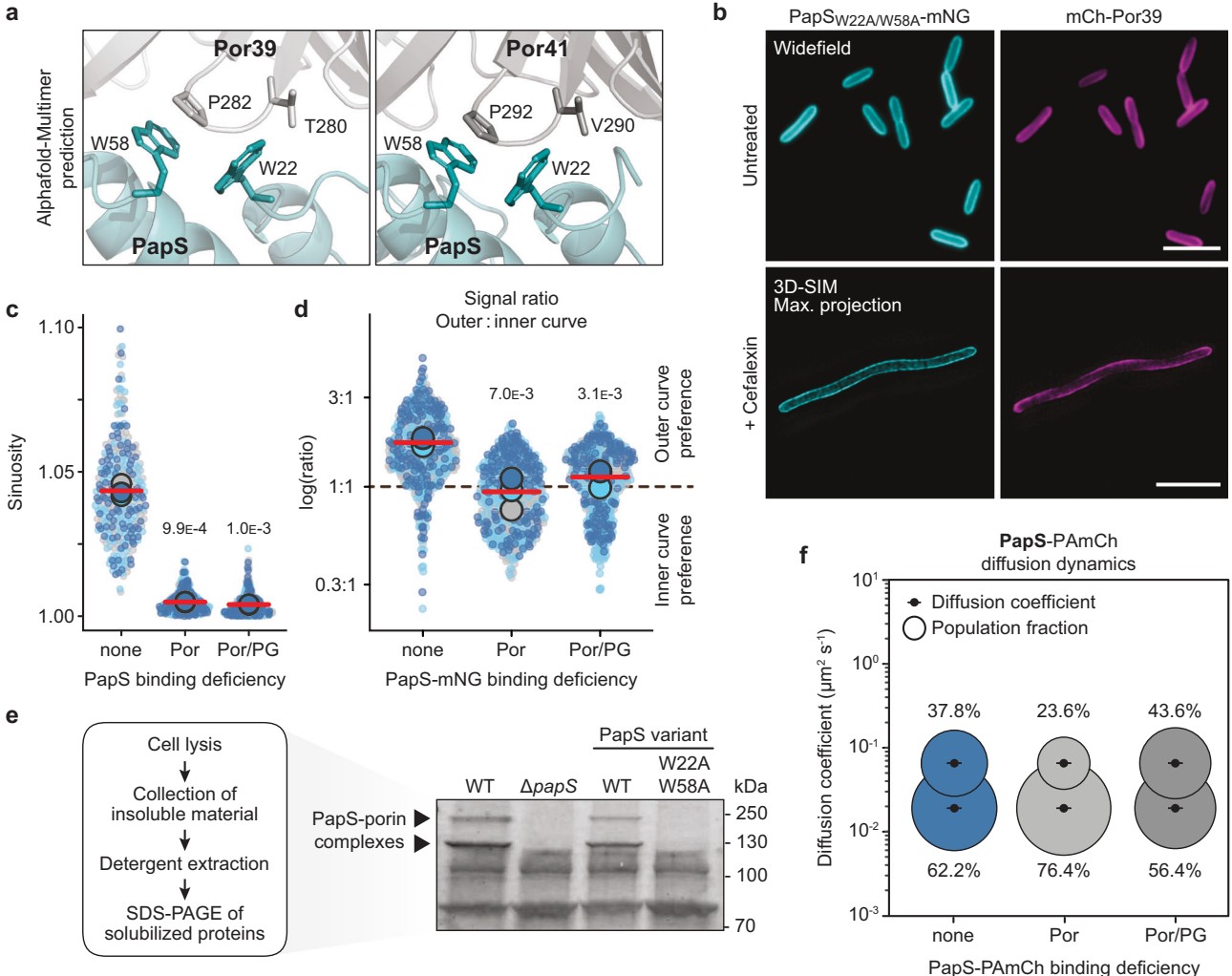

**Fig. 6 | Porin binding is required for PapS localization. a** Model of the interaction interfaces of *R. rubrum* PapS (turquoise) and Por39 (light grey) or Por41 (dark grey), generated with AlphaFold-Multimer[48]. Amino acids predicted to be involved in the interaction are shown in stick representation and labeled. **b** Colocalization of PapS$_{W22A/W58A}$-mNG with mCh-Por39 in the Δ*papS* background. Cells (SP146) were cultivated in the absence (untreated) or presence (+) of cefalexin prior to imaging. The images show representative cells analyzed by widefield epifluorescence microscopy (untreated) and 3D-SIM (+ Cefalexin) (*n* = 3 biological replicates). Bar: 5 μm. The area overlap of the PapS$_{W22A/W58A}$-mNG and mCh-Por39 signals in the 3D-SIM images is 60.7% (see also Supplementary Data 4). **c** Superplots showing the distribution of cell sinuosities in populations of Δ*papS* cells producing wild-type PapS (JR54), the porin-binding-deficient variant PapS$_{W22A/W58A}$ (SP186), or the porin- and peptidoglycan-binding-deficient variant PapS$_{W22A/W58A/R223A}$ (SP187) from a low-copy plasmid. The data are presented as described for Fig. 1f (*n* = 100 cells per replicate). The statistical significance (*p* values) of differences between cells producing the wild-type and the mutant proteins is indicated (unpaired two-sided Welch's *t*-test). **d** Superplots showing a quantification of the outer-versus-inner-curve mNG signal ratios in Δ*papS* cells producing wild-type PapS-mNG (SP136), the porin-binding-deficient variant PapS$_{W22A/W58A}$-mNG (SP197) or the

porin- and peptidoglycan-binding-deficient variant PapS$_{W22A/W58A/R223A}$-mNG (SP174) from a low-copy plasmid. The data are presented as described for Fig. 1f (*n* = 100 cells per replicate). The results for wild-type PapS-mNG are replotted from Fig. 3e as a reference. The statistical significance (*p* values) of differences between cells producing the wild-type and the mutant proteins is indicated (unpaired two-sided Welch's *t*-test). **e** SDS gel showing the presence or absence of porin-PapS complexes in Triton extracts of *R. rubrum* wild-type (S1) and Δ*papS* (JR52) cells and of cells producing wild-type PapS (JR54) or PapS$_{W22A/W58A}$ (SP186) from a low-copy plasmid. **f** Bubble plots showing the apparent single-molecule diffusion rates of different PapS-PAmCh variants in the Δ*papS* background. Cells producing wild-type PapS-PAmCh (SP165), the porin-binding-deficient variant PapS$_{W22A/W58A}$-PAmCh (SP168), or the porin- and peptidoglycan-binding-deficient variant PapS$_{W22A/W58A/R223A}$-PAmCh (SP175) from a low-copy plasmid were subjected to SPT analysis. The results for cells producing wild-type PapS-PAmCh are replotted from Fig. 3f as a reference. The diffusive and immobile populations have mean apparent diffusion coefficients of 0.0657 μm$^2$ s$^{-1}$ and 0.0191 μm$^2$ s$^{-1}$, respectively. The number of cells analyzed is given in Supplementary Data 3. Source data are provided as a **Source Data** file.

interaction interface. For this purpose, we predicted the molecular structure of a binary Por41-PapS complex using AlphaFold-Multimer[48]. The model obtained suggested that the interaction between the two proteins involved two surface-exposed tryptophan residues of PapS (W22, W58) that formed a hydrophobic pocket accommodating a conserved hydrophobic loop at the periplasmic face of Por41 (Fig. 6a and Supplementary Fig. 7). To verify the predicted interface, we generated variants of PapS(-mNG) carrying substitutions in either or both of the tryptophan residues and tested

their functionality by complementation analysis in vivo. Single substitutions (W22A, W58A) did not have any effect on the localization of the fusion protein or its ability to mediate cell curvature (Supplementary Fig. 9a, b). However, the combination of the two exchanges completely abolished the helical localization pattern of PapS-mNG, accompanied by a loss of curved cell morphology, while the localization of mCh-Por39 was unaffected in this background (Fig. 6b–d, Supplementary Figs. 9b–d and 10a–d, Supplementary Movies 12 and 13 and Supplementary Data 4). These findings indicate

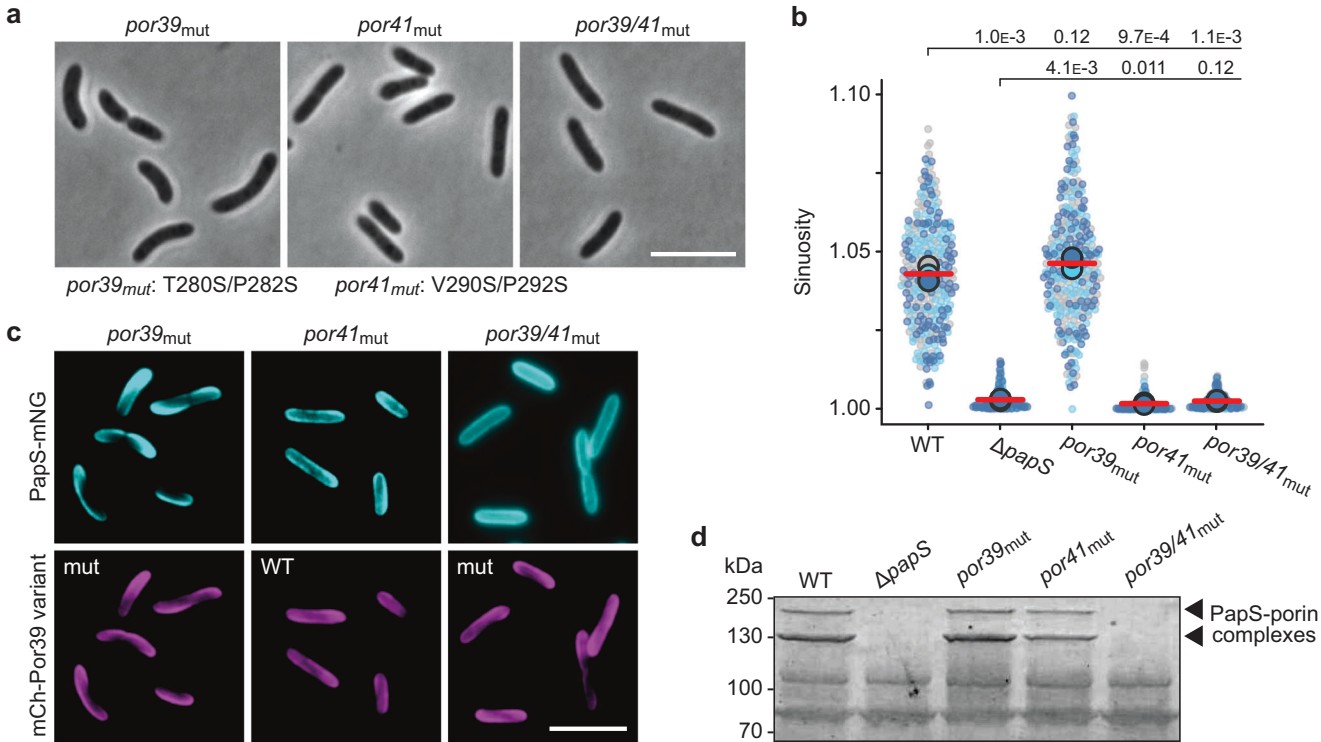

**Fig. 7 | Both Por39 and Por41 contribute to PapS localization. a** Morphology of *R. rubrum* cells producing mutant porin variants with exchanges in the predicted PapS binding site. Shown are *por39mut* (SP235), *por41mut* (SP234), and *por39mut por41mut* (SP242) cells, visualized by phase-contrast microscopy. Bar: 5 μm. **b** Superplots showing the distribution of cell sinuosities in populations in the indicated *R. rubrum* strains. The data are presented as described for Fig. 1f. The statistical significance (*p* value) of differences between strains is indicated

(unpaired two-sided Welch's *t*-test). **c** Colocalization of PapS-mNG and the indicated mCh-Por39 variants in the *por39mut* (LMS23), *por41mut* (LMS24) and *por39/41mut* (SP243) backgrounds. Shown are representative cells (*n* = 3 biological replicates), analyzed by widefield epifluorescence microscopy. Bar: 5 μm. **(d)** SDS gel showing the presence or absence of porin-PapS complexes in Triton extracts of the indicated *R. rubrum* strains. Source data are provided as a **Source Data** file.

an essential role of W22 and W58 in the association of PapS with the two outer-membrane porins. Moreover, they strongly support the hypothesis that the porin-PapS interaction is critical for PapS localization and that the helical arrangement of PapS plays a decisive role in cell curvature.

To verify that the defects in the function of PapS were indeed caused by a disruption of its interaction with Por39/41, we devised a biochemical approach to directly assay cells for the presence of porin-PapS complexes. To this end, we isolated cell envelopes from strains producing wild-type PapS or a mutant variant lacking the two conserved tryptophan residues (Supplementary Fig. 10a), extracted membrane proteins by detergent treatment, and then separated them by gel electrophoresis. Due to their high abundance (compare Fig. 1a) and stability[31], porin-PapS complexes were readily identified as prominent, slow-migrating bands in extracts of cells producing wild-type PapS (Fig. 6e and Supplementary Data 5). Cells producing the double-mutant protein, by contrast, lacked these bands and showed a protein pattern similar to that of the ΔpapS strain, confirming a key role of W22 and W58 in porin binding.

Notably, single-particle tracking analysis of mutant PapS-PAmCh variants (Supplementary Fig. 10e, f) revealed that the defect in porin binding strongly decreased the diffusional motility of the respective fusion protein (Fig. 6f and Supplementary Data 3), to an extent even greater than the previously mentioned loss of the N-terminal lipid anchor (compare Fig. 3f). Additional deletion of the OmpA-like pepti-doglycan-binding domain (Fig. 6c, d, f and Supplementary Movie 14) reversed this effect and led to a significant recovery of protein mobility, although the proportion of mobile molecules remained lower than that observed for an analogous fusion with an intact porin binding site (compare Fig. 3f). The observed (low) mobility of PapS within the outer

membrane thus depends both on porin binding and the presence of a lipid anchor.

Next, we aimed to verify the PapS-binding site of the porins. To this end, we generated strains producing porin variants with substitutions in the predicted interaction determinants (Supplementary Fig. 11a, b), including T280S and P282S for Por39 and V290S and P292S for Por41 (Fig. 6a). The mutation of Por39 alone did not have a significant effect on cell shape (Fig. 7a, b and Supplementary Fig. 11c) nor did it affect the localization of PapS-mNG (Fig. 7c). Analogous substitutions in Por41, by contrast, completely abolished cell curvature (Fig. 7a, b and Supplementary Fig. 11c). Interestingly, PapS-mNG still formed helical structures in this background, although the pattern appeared to be less defined. However, when both porins were mutated, the distribution of the fusion protein became entirely uniform, while porin localization remained unchanged (Fig. 7c). Consistent with these results, the analysis of membrane extracts showed that the levels of porin-PapS complexes were not affected by substitutions in Por39. However, they were markedly reduced upon mutation of Por41 and dropped below the detection limit in cells carrying substitutions in both porin paralogs (Fig. 7d). Together, these findings confirm the predicted porin-PapS interfaces. Moreover, they demonstrate that both porins contribute to PapS localization, even though the more abundant Por41 protein has a dominant role in this process.

## PapS mediates elongasome entrapment at the outer curve of the cell

The results obtained showed that the porin-mediated formation of helical PapS ribbons was critical for the establishment of cell curvature in *R. rubrum*. However, it remained to be clarified how the assembly of

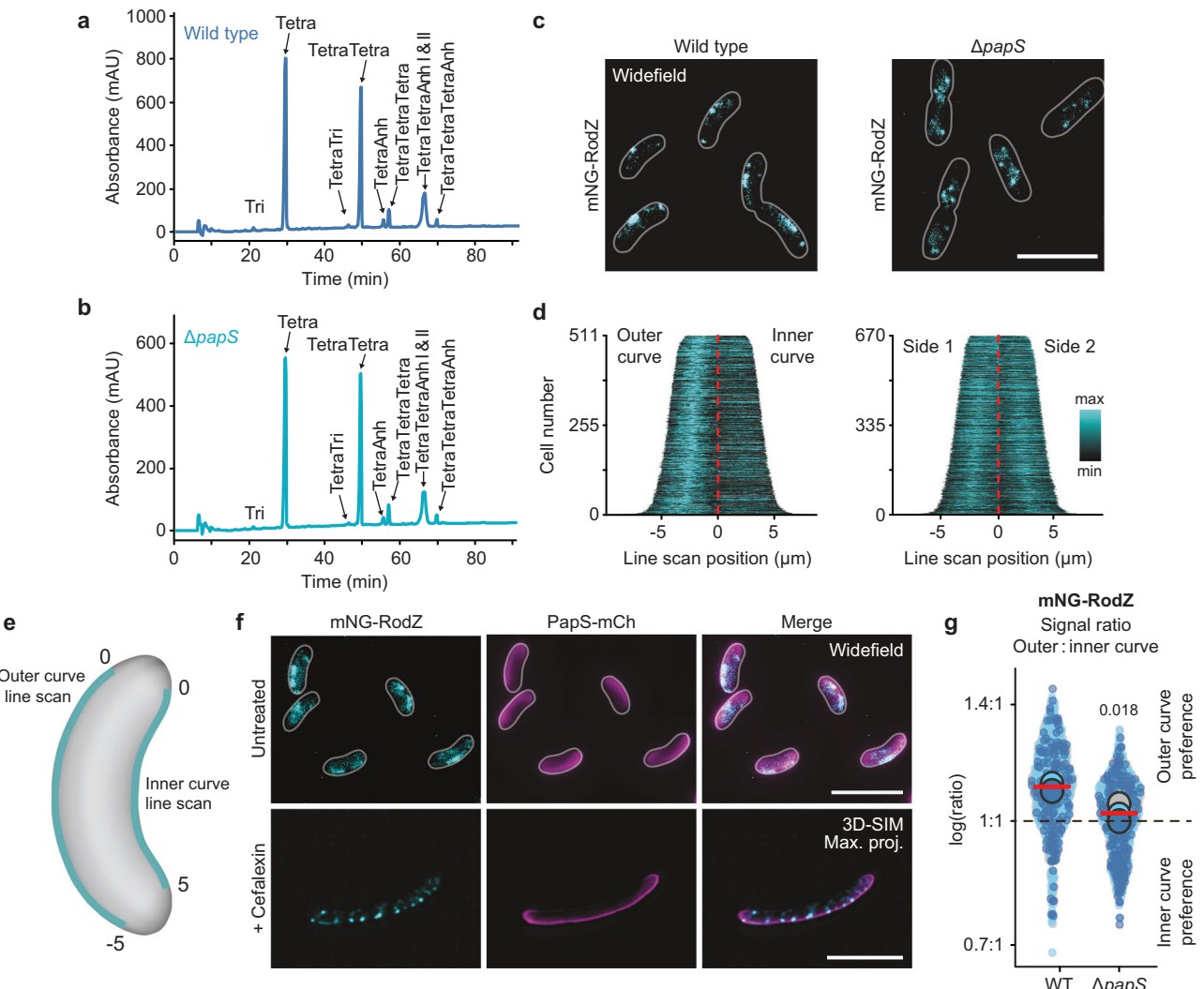

**Fig. 8 | PapS promotes the enrichment of elongasome complexes at the outer curve. a, b** Muropeptide profiles of *R. rubrum* wild-type (S1) and Δ*papS* (JR52) cells. The major muropeptide species are annotated. "Tetra" stands for N-acetyl-glucosamine-N-acetylmuramic acid tetrapeptide. "Anh" indicates Tetra derivatives containing a 1,6-anhydro-N-acetylmuramic acid moiety. **c** Localization of mNG-RodZ in the wild-type (SP160) and Δ*papS* (SP162) backgrounds by widefield epi-fluorescence microscopy. Bar: 5 μm. Shown are representative cells (*n* = 3 biological replicates). **d, e** Demographs showing the distribution of mNG-RodZ fluorescence along the outer and inner curve in populations of wild-type (SP160; *n* = 511 cells) and Δ*papS* (SP162; *n* = 670 cells) cells. The outer- and inner-curve fluorescence intensity profiles obtained for individual cells were concatenated (as exemplified in (**e**)) and stacked on top of each other according to their combined length. **f** Colocalization of PapS-mCh with mNG-RodZ in the wild-type background (SP163). Shown are representative cells (*n* = 3 biological replicates) cultivated in the absence (untreated) or presence (+) of cefalexin and analyzed by widefield fluorescence microscopy or 3D-SIM, respectively. Bar: 5 μm. The area overlap of the PapS-mCh and mNG-RodZ signals in the 3D-SIM images is 57.9% (see also Supplementary Data 4). **g** Superplots showing the outer-versus-inner-curve mNG-RodZ signal ratios in the wild-type (SP160) and Δ*papS* (SP162) backgrounds. The data are presented as described in Fig. 1f. The statistical significance (*p* value) of differences between strains is indicated (unpaired two-sided Welch's *t*-test). Source data are provided as a **Source Data** file.

these structures translated into asymmetric peptidoglycan remodeling (compare Fig. 1g, h), thereby giving rise to a helical cell shape. Our calculations showed that a small (~15%) fractional increase in the extent of cell wall growth at the outer curve was sufficient to produce the typical curvature observed for *R. rubrum* cells (Supplementary Note 1). To directly visualize peptidoglycan biosynthesis, we treated cells with a fluorescent D-alanine analog (HADA) that specifically labeled newly incorporated cell wall material (Supplementary Fig. 12a). Apart from strong signals at the cell division sites, we observed prominent arc- or ring-like patterns that spanned the circumference of the cell along the longitudinal side walls, reminiscent of the movement pattern reported for elongasome complexes[49,50]. However, there was no significant difference in the integrated fluorescence intensities between the inner and outer curve (Supplementary Fig. 12b), suggesting that the expected increase in HADA incorporation at the outer curve was below the

detection limit or compensated by a higher turnover of labeled material in this cellular region.

In a different approach to shed light on the mechanism of PapS-mediated cell curvature, we first determined whether the activity of PapS affected the global composition of the *R. rubrum* cell wall. This analysis showed that the muropeptide profile of the Δ*papS* mutant was highly similar to that of the wild-type strain (Fig. 8a, b and Supplementary Data 6), suggesting that PapS may not recruit curvature-specific enzymes that alter the structure of the cell wall at the outer curve, but rather modulate the activity of the standard cell elongation machinery. We reasoned that the association of PapS with both the outer membrane and the peptidoglycan layer could, for instance, affect the width of the periplasmic space and thus modulate the accessibility of the cell wall biosynthetic machinery to outer membrane-bound regulatory factors[51–53]. However, cryo-electron

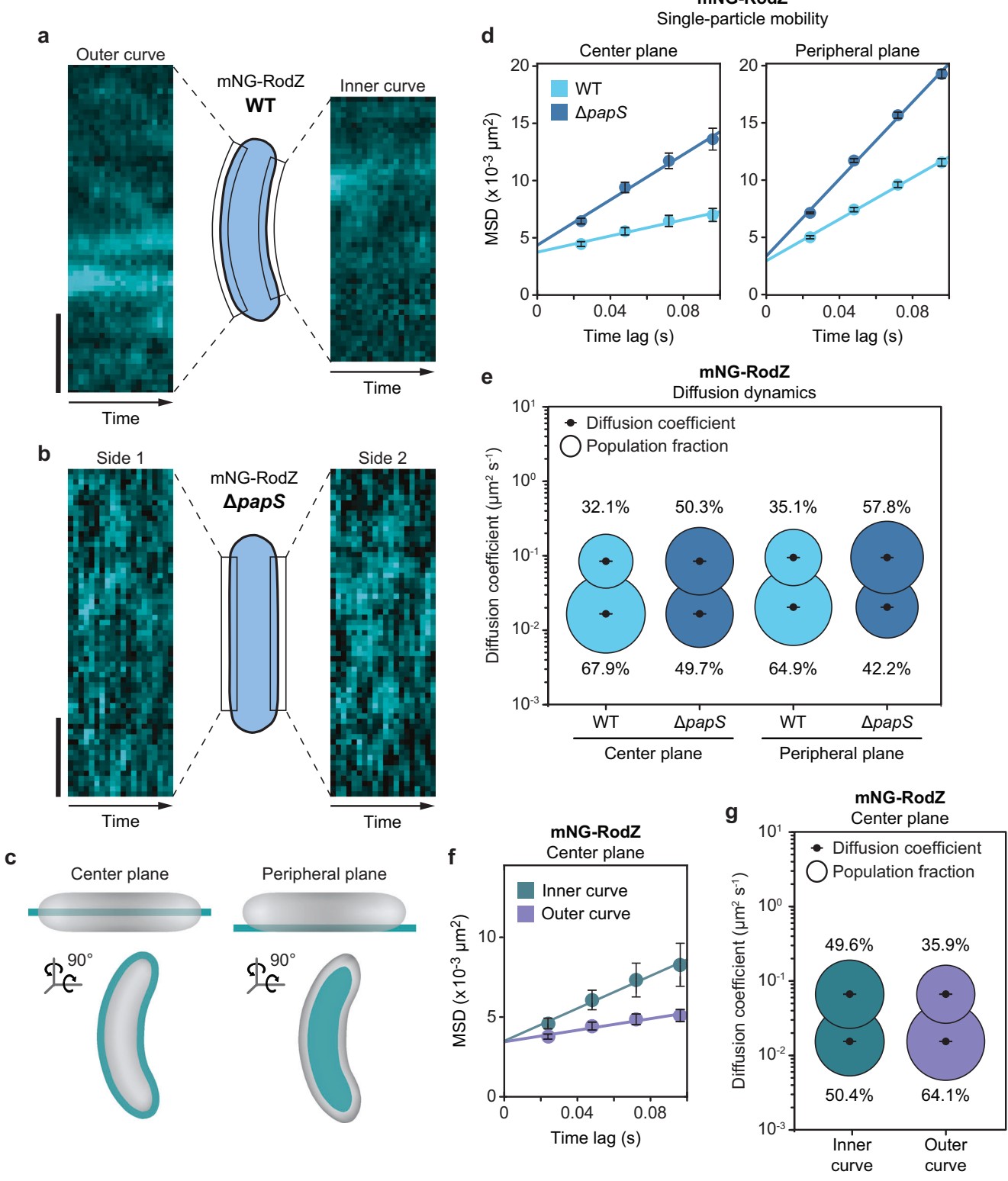

tomography analysis did not reveal any difference in the inter-membrane spacing at the inner and outer curve of *R. rubrum* wild-type cells (Supplementary Fig. 13). Therefore, we went on to investigate whether PapS could have a direct effect on elongasome movement. To this end, we aimed to construct strains in which individual elongasome components were fluorescently tagged with mNG. While most of the fusion constructs turned out to be non-functional, it was possible to obtain a strain producing a tagged variant of the bitopic

membrane protein RodZ (Supplementary Fig. 14a–c), an integral part of the elongasome linking the cytoplasmic MreB cytoskeleton with the membrane-bound components of the complex[54–56]. Microscopic analysis revealed that mNG-RodZ was strongly enriched at the outer curve of the cell, where it colocalized with the PapS structures (Fig. 8c–g, Supplementary Fig. 14a–c, Supplementary Movie 15 and Supplementary Data 4). In the Δ*papS* background, this pattern was completely abolished and mNG-RodZ foci were evenly distributed

**Fig. 9 | PapS reduces the motion of elongasome complexes. a** Kymographs showing the distribution of mNG-RodZ fluorescence along the outer and inner curve of the cell over time (1-sec intervals over a total of 20 sec) in the wild-type background (SP160). Shown is a representative cell (*n* = 3 biological replicates). Each column of pixels corresponds to one time point. Bar: 1 μm. **b** Kymographs showing the distribution of mNG-RodZ fluorescence along opposite sides of the cell over time (1-sec intervals over a total of 20 sec) in the Δ*papS* background (SP162), generated as in (**a**). **c** Schematic illustrating the positions of the center and peripheral planes of the cell. **d** Mobility of mNG-RodZ in the wild-type (SP160) and Δ*papS* (SP162) backgrounds. Single mNG-RodZ molecules were tracked in the center plane or the peripheral plane of the cell (see (**c**)) by total internal reflection fluorescence (TIRF) microscopy and subjected to mean square displacement (MSD) analysis. Data represent the mean (±SD) of all tracks analyzed. **e** Bubble plots showing the single-molecule diffusion rates of mNG-RodZ

in the center plane or peripheral plane of the cell in the wild-type (SP160) and Δ*papS* (SP162) backgrounds. The diffusive and immobile populations have mean apparent diffusion coefficients of 0.0884 μm² s⁻¹ and 0.0174 μm² s⁻¹ (center plane) or 0.0992 μm² s⁻¹ and 0.0214 μm² s⁻¹ (peripheral plane), respectively. The number of cells analyzed for each condition in (**d**, **e**) is given in Supplementary Data 3. **f** Mobility of mNG-RodZ at the inner and outer curve, respectively. The plots were generated by reanalyzing the dataset obtained for mNG-RodZ in the wild-type background (SP160), imaged in the center plane of the cell as described for (**d**, **e**). Data represent the mean (±SD) of all tracks analyzed. **g** Bubble plots showing the apparent single-molecule diffusion rates of mNG-RodZ at the inner and outer curve, respectively. The diffusive and immobile populations have mean apparent diffusion coefficients of 0.0663 μm² s⁻¹ and 0.0154 μm² s⁻¹, respectively. The number of cells analyzed for each condition in (**f**, **g**) is given in Supplementary Data 3.

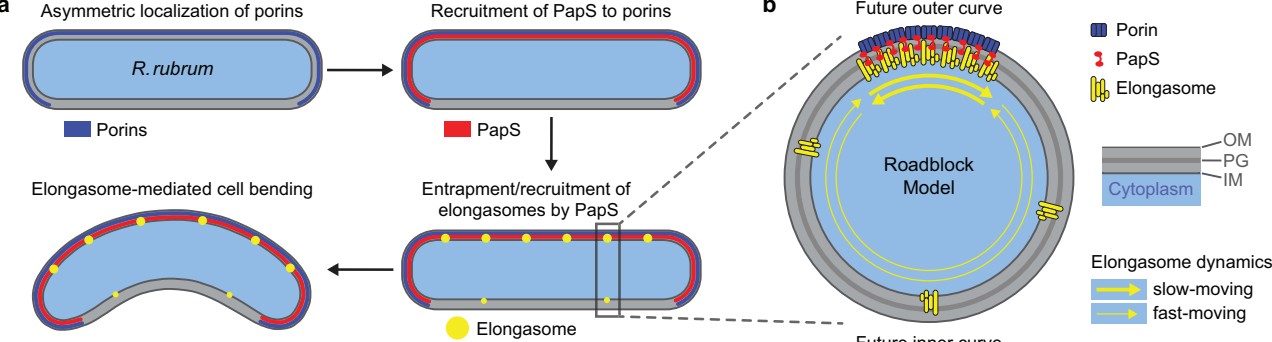

**Fig. 10 | Model of the establishment of cell curvature in *R. rubrum*. a** Proposed mechanism underlying the PapS-dependent asymmetry of peptidoglycan biosynthesis in *R. rubrum* cells. The two porins Por41 and Por39 are arranged in an asymmetric, helical pattern and recruit PapS through interactions with its N-terminal porin-binding interface. The resulting porin-PapS assemblies entrap elongasome complexes and thus promote elevated longitudinal growth of the peptidoglycan sacculus in the cell envelope regions they occupy, thereby inducing

cell bending. **b** Scheme illustrating the roadblock model of PapS function. The helical porin-PapS assemblies are tightly associated with the outer membrane and the peptidoglycan layer. Their high density creates a physical barrier that cages elongasome complexes by hindering their motion and thus decreasing their processivity, promoting frequent changes in their direction of movement. Sporadically, elongasome complexes escape this barrier, allowing cell elongation to occur in other cellular regions, albeit at a reduced rate.

throughout the cell envelope, suggesting that the presence of PapS affects the behavior of elongasome complexes.

To clarify the impact of PapS on elongasome assembly and movement, we followed the dynamics of mNG-RodZ localization by time-lapse microscopy. In the wild-type background, mNG-RodZ foci assembled preferentially at the outer curve, where they persisted over prolonged periods of time, either staying in place or slowly moving along the longitudinal axis (Fig. 9a, Supplementary Fig. 15a, Supplementary Movies 16 and 17). In the straight Δ*papS* mutant, by contrast, the foci were considerably more dynamic and evenly distributed between the two sides of the cell (Fig. 9b, Supplementary Fig. 15b, Supplementary Movies 18 and 19), suggesting that PapS structures promote the formation of more stable but less mobile elongasome complexes. Consistent with this notion, single-particle tracking studies showed that the absence of PapS led to a considerable increase in the proportion of diffusive mNG-RodZ molecules, pointing to shorter elongasome lifetimes (Fig. 9c−e and Supplementary Data 3). Notably, this effect was not only observed for molecules in the center plane of the cell, which includes the PapS ribbons, but also in the peripheral planes, which are less densely populated by PapS (compare Fig. 2f). PapS thus appears to affect the dynamics of elongasome complexes throughout the cell envelope, although its influence may be most pronounced within the compact PapS assemblies at the outer curve. A quantification of the spatial distribution of single-particle tracks in the center plane confirmed that both fast- and slow-moving mNG-RodZ molecules were enriched at the outer curve in wild-type cells, whereas they were evenly distributed to both sides of the cell in

the Δ*papS* mutant (Supplementary Fig. 15c). Moreover, when separating the single-particle tracks obtained in wild-type cells according to their subcellular location, we found that the overall mobility of mNG-RodZ molecules at the outer curve was significantly lower than that at the inner curve (Fig. 9f, g). Molecules in the peripheral plane, by contrast, did not show any localization bias in either of the two strain backgrounds (Supplementary Fig. 15d). Together, these findings strongly suggest that the curved morphology of *R. rubrum* cells is based on a spatial bias in the formation and localization of elongasome complexes, established by PapS-mediated elongasome entrapment at the outer cell curve.

## Discussion

Many bacteria have evolved curved or spiral morphologies, which confer a significant fitness advantage in the environments they inhabit[1–3]. The underlying mechanisms are highly diverse, although most of the systems involved rely on protein polymers that localize to the inner or outer curve of the cell to mediate local changes in the cell elongation rate[17]. Despite the considerable progress made, the molecular links between these curvature determinants and the peptidoglycan biosynthetic machinery are still incompletely understood. In this study, we identify a new type of curvature-inducing system, relying on helically organized outer-membrane protein assemblies that modulate the dynamics of the elongasome complex to promote peptidoglycan biosynthesis in a defined longitudinal zone of the cell cylinder. These findings not only shed light on the mechanistic basis of cell morphogenesis in a large and abundant group of bacteria but also

reveal an unprecedented degree of spatial organization in a bacterial outer-membrane proteome. Moreover, they identify a key role of outer-membrane protein patterning in the spatiotemporal regulation of inner-membrane protein dynamics, thus adding a new facet to the diverse repertoire of regulatory processes in bacterial cell biology.

Cell curvature in *R. rubrum* critically depends on the porins Por41 and Por39, whose helical arrangement in the outer membrane provides the spatial landmark guiding asymmetric cell wall growth. Notably, while cell curvature per se is dispensable for cell viability, both porins are essential, suggesting that they are moonlighting proteins that still function as channels mediating solute exchange across the outer membrane but have evolved an additional, physiologically unrelated role in cell morphogenesis. This functional dichotomy may also explain why it was not possible to fluorescently tag the major porin Por41. Previous work has shown that Por41/39 assemble into homo- or heterotrimeric complexes[31], which together form the conducting channel. Due to its significantly higher abundance (Fig. 1a), Por41 will predominantly form homomeric complexes, whose assembly may be sterically hindered by the large fluorescent protein tag, resulting in a severe defect in porin function. The majority of Por39 molecules, by contrast, are likely to be incorporated into heterotrimers, as suggested by the dominant effect of mutations in Por41 on Por39 localization. In their case, the tags may be more loosely packed and thus leave porin function unaffected. The mechanism mediating the helical localization of porin complexes is still unclear. It is conceivable that porin trimers can self-assemble into helical superstructures, in a process mediated either by direct interactions between porin trimers or by as-yet unknown adapter proteins. Alternatively, porin subunits could be secreted and inserted into the outer membrane by a helically organized machinery and then be retained at the site of insertion due to the relatively static nature of the asymmetric outer-membrane bilayer[57]. Notably, recent work has shown that outer-membrane proteins are mostly integrated at sites of ongoing peptidoglycan biosynthesis[58]. The establishment of porin assemblies could therefore be facilitated by a self-organizing process in which porin-PapS complexes promote cell wall biosynthesis in their vicinity, thereby in turn increasing the rate of porin insertion at these sites. The resulting local increase in cell curvature could further bias the porin distribution if porin complexes, for instance, formed intrinsically curved oligomers that preferentially assembled and/or localized in regions of positive Gaussian curvature. This process could potentially be facilitated by the enrichment of certain lipid species or lipid microdomains in these curved regions[59].

The central effector of the curvature-inducing system is PapS, an outer-membrane lipoprotein that bridges the helical porin assemblies and the peptidoglycan layer. Our work indicates that PapS does not form stable polymeric structures but dense arrays of individual monomers (Fig. 2f–h) that act as roadblocks slowing down the circumferential motion of elongasome complexes. Since elongasome movement is required for peptidoglycan biosynthesis, a simple stalling mechanism would not lead to increased incorporation of new cell wall material. We, therefore, propose that PapS ribbons act as cages that transiently entrap elongasomes as they pass through them, only allowing short-distance back-and-forth movements in between adjacent PapS roadblock complexes (Fig. 10). During each helical turn, elongasomes thus incorporate more peptidoglycan in the regions occupied by PapS than in the remaining parts of the cell envelope, leading to uneven cell elongation. This caging mechanism may be sufficient to explain both the establishment and the maintenance of cell curvature. In straight cells, opposite sides of the cell body show the same density of elongasome tracks. The increased incorporation of peptidoglycan at PapS ribbons thus results in a relative increase in the rate of cell elongation per unit surface area in a helical longitudinal sector of the cell, resulting in cell curvature. As the cell body bends, the density of elongasome tracks at the outer curve becomes increasingly

lower, while that at the inner curve remains unchanged. As a consequence, the elongation rates at the two curves gradually converge, until the cell reaches a steady state in which both sides grow at the same rate, maintaining a constant degree of curvature. The central role of outer-membrane porins in the control of elongasome dynamics reverses the typical inside-to-outward hierarchy in the regulation of cell envelope biogenesis whereby cytoplasmic protein polymers control the localization of enzyme complexes in the cell membranes and the periplasmic space. This finding adds to previous work that has revealed roles for the outer membrane and outer-membrane lipoproteins in rod-shape determination[60] and the temporal regulation of peptidoglycan biosynthesis, respectively, in *E. coli*[51–53] and thus emphasizes the emerging significance of the outer membrane in cell wall biology.

The confinement of elongasome complexes by roadblocks also contributes to cell shape determination in other proteobacterial species. For instance, cytoplasmic bactofilin polymers were shown to generate barriers that limit elongasome movement to certain regions of the cell envelope, thereby facilitating the formation of cellular extensions such as stalks and buds[61,62]. Notably, bactofilin has recently also been identified as a modulator of cell shape in *R. rubrum*, where it forms complexes with a peptidoglycan hydrolase that stimulate peptidoglycan biosynthesis at the inner curve, thus antagonizing the curvature induced by the PapS system[61]. This bipartite system is superficially reminiscent of the curvature-inducing morphogenetic machinery in *H. pylori*, which involves the recruitment of bactofilin-peptidoglycan hydrolase complexes to the outer curve as well as the enrichment of the elongasome component MreB at the inner curve, mediated by thus-far unknown localization mechanisms. Thus, both the modulation of elongasome movement and site-specific peptidoglycan hydrolysis emerge as common themes in the establishment of complex bacterial morphologies. It will be interesting to determine whether these mechanistic principles could also apply to other curvature-inducing protein polymers reported to date.

In terms of its physiological properties, PapS shares similarities with proteins such as *E. coli* Pal, OmpA, and Lpp, all of which are highly abundant in the cell and ensure cell envelope integrity by stably connecting the outer membrane and the peptidoglycan layer[40,63,64]. Given the fitness defect observed for the ΔpapS mutant, PapS could have a similar role in *R. rubrum*, but cryo-electron tomography did not reveal any obvious changes in the integrity of the outer membrane or the width of the periplasmic space (Supplementary Fig. 13). Compared to OmpA from *A. baumannii*, PapS has a lower affinity for mDAP and may therefore be less tightly associated with the cell wall. This property could potentially make it less suitable as an architectural protein, although the high abundance and dense packing of PapS in the outer membrane may still allow efficient, albeit more transient, peptidoglycan binding. Faster dissociation kinetics may ensure that mDAP residues in uncrosslinked peptide side chains remain available for transpeptidation. Moreover, they may help to keep porin-PapS complexes in a dynamic state that facilitates the escape of trapped elongasomes. Interestingly, many members of the *Rhodospirillales* possess more than one PapS homolog. It is conceivable that the functions of these different PapS proteins have diverged during evolution, with some acting as cell shape determinants and others as stabilizing factors ensuring cell envelope integrity. Cells could also contain multiple curvature-inducing PapS orthologs that differ in their porin- and peptidoglycan-binding affinities and thus have distinct roadblock activities, enabling an adjustment of cell curvature to variations in the environment they inhabit.

It will be interesting to clarify the conservation of the PapS system and the role of PapS homologs in the *Rhodospirillales* and to see if analogous systems are at work to generate complex cell shapes in other lineages. Moreover, it will be revealing to determine whether outer-membrane protein patterning is a more common phenomenon

in *R. rubrum* or bacteria in general. Our work demonstrates that the study of bacterial morphogenesis continues to provide a rich source of insight into the molecular principles of cellular organization in bacteria, and intensive efforts are required to fully unravel the systems producing the large variety of morphologies known to date.

# Methods

## Growth conditions

*R. rubrum* S1 (ATCC 11170)[28–30] and its derivatives were grown aerobically in the dark at 28 °C on Tryptic Soy (TS) medium (Becton Dickinson, USA) plates or in liquid TS medium shaking at 210 rpm in Erlenmeyer flasks, unless indicated otherwise. Media were supplemented with 30 μg ml$^{-1}$ kanamycin when appropriate, to ensure the maintenance of integrative or replicating plasmids. Cell filamentation was induced by treating cells with 5 μg ml$^{-1}$ cefalexin for 6 h before imaging. To analyze cell growth, *R. rubrum* cells were grown to the exponential phase, diluted with fresh medium to an optical density at 600 nm (OD$_{600}$) of 0.01, and transferred into 24-well polystyrene microtiter plates (Becton Dickinson Labware, USA). The increase in cell mass was then followed at 28 °C under double-orbital shaking in an EPOCH 2 microplate reader (BioTek, USA) by measuring the OD$_{600}$ at 10 min intervals.

*E. coli* strains were cultivated at 37 °C on LB plates or in liquid LB medium shaking at 210 rpm. For plasmid-bearing strains, antibiotics were added at the following concentrations (μg ml$^{-1}$ in liquid/solid medium): kanamycin (30/50), ampicillin (50/200). To grow *E. coli* WM3064, media were supplemented with 2,6-diaminopimelic acid (DAP) at a final concentration of 300 μM.

## Plasmid and strain construction

The bacterial strains, plasmids, and oligonucleotides used in this study are listed in Supplementary Tables 1-4. The sequences of *papS*, *por39*, and *por41* were retrieved from the genome of *R. rubrum* S1[33]. *E. coli* TOP10 (Thermo Fisher Scientific, USA) was used as a host for cloning purposes. All plasmids were verified by DNA sequencing.

*R. rubrum* was transformed by conjugation using *E. coli* strain WM3064 as a donor[65]. Gene replacement was achieved by double-homologous recombination using the counter-selectable *sacB* marker[66]. Proper chromosomal integration or gene replacement was verified by colony PCR. Details on the construction of strains and plasmids are given in Supplementary Tables 1 and 3.

## Ampicillin resistance assay

Samples (5 μl) of an exponentially growing (OD$_{600}$ ≈ 0.4) *R. rubrum* culture were spread on two TS agar plates containing no supplement or 200 μg ml$^{-1}$ ampicillin, respectively. The plates were incubated for 7 days at 28 °C and imaged using a ChemiDoc MP imaging system (BioRad, USA).

## Gentle cell lysis

Gentle cell lysis was performed as described previously[20]. *R. rubrum* cells were grown to exponential phase and subsequently placed on TS agarose-pads containing 1% agarose, 50 mM EDTA, 100 μg ml$^{-1}$ polymyxin B, 10 mM lysozyme and 50 μg ml$^{-1}$ fosfomycin. Cell lysis was followed by phase contrast and fluorescence microscopy at 1 min intervals for 1 h.

## Phase-contrast and widefield-epifluorescence imaging

Cells were grown to the exponential phase in the appropriate medium, transferred onto 1% agarose pads and imaged with a Zeiss Axio Imager.M1 microscope equipped with a Zeiss Plan Apochromat 3 100/1.40 Oil DIC objective and a pco.edge 3.1 sCMOS camera (PCO) or with a Zeiss Axio Imager.Z1 microscope equipped with a 100x Plan-Apochromat Oil Ph3 M27 objective and a pco.edge 4.2 sCMOS camera (PCO, Germany). For epifluorescence detection, an X-Cite 120PC metal halide light source (EXFO, Canada) and ET-YFP, ET-DAPI or ET-TexasRed filter cubes (Chroma, USA) were used. Images were recorded with VisiView 3.3.0.6 (Visitron Systems, Germany) and processed with Fiji 1.53[67] and Adobe Illustrator CS6 (Adobe Systems, USA).

Cell sinuosity was measured using the ImageJ addon MicrobeJ[68]. Three independent sets (n = 100) of cells were analyzed per strain for quantification. Results were displayed using the SuperPlotsOfData application[69]. An unpaired Welch's *t*-test, as implemented in the SuperPlotsOfData application, was used to identify significant differences on the cell sinuosity between conditions. In order to characterize the ratio of total fluorescence intensity in the outer and the inner curve of cells, a custom-made algorithm was established in Fiji 1.53[67] and R[70] as described previously[71]. In brief, the outlines of single cells were determined in phase contrast micrographs, and the respective curved centerline was constructed computationally. Based on the intersections of these two traits, line-scans were applied on the inner and the outer cell-half in the corresponding fluorescence images. From these measurements, the mean/median values were calculated, and the ratio of the outer versus the inner cell curvature was determined. Experiments were performed in triplicate and the number of cells analyzed is given in the **Source Data** file. The ratio measurements were visualized and analyzed using the SuperPlotsOfData application[69].

## HADA staining

HADA staining was performed as decribed previously[72]. *R. rubrum* cells were grown to an OD$_{600}$ of 0.3, treated with 1 mM HADA for 30 s, harvested by centrifugation (2 min, 9400 ×g) at room temperature, and resuspended in 1 ml of 70% (v/v) ice-cold ethanol. After incubation on ice for 10–15 min and three washes in 1 ml PBS (137 mM NaCl, 2.7 mM KCl, 10 mM Na$_2$HPO$_4$, 1.8 mM KH$_2$PO$_4$, pH 7.4), the fixed cells were resuspended in 20 μl of PBS, transferred to pads made of 1% agarose in PBS and imaged using a DAPI filter set.

## Fluorescence-recovery-after-photobleaching (FRAP) analysis

FRAP analysis was performed with an Axio Observer.Z1 inverted microscope (Carl Zeiss, Germany), equipped with a 2D-VisiFRAP multi-point FRAP module (Visitron Systems, Germany) and a 561 nm solid-state laser. After the acquisition of 10 pre-bleach images, regions of interest (ROI; 8 × 8 pixels) were bleached with 64-ms pulses (1 ms per pixel) at 100% laser intensity, and signal recovery was followed for 15 min, with images taken at 10-sec intervals. To analyze the data, the integrated fluorescence intensities of the ROI and the whole cell were determined for each time point $t$ and corrected for background fluorescence, yielding the value $R_t$ and $W_t$ respectively. The same analysis was performed on each the 10 pre-bleach images, and the values obtained were averaged to obtain the corresponding mean integrated pre-bleach fluorescence intensities $R_{PB}$ and $W_{PB}$. Subsequently, the normalized integrated fluorescence intensity of the ROI $F_t$ was determined using the equation

$$F_t = \left(\frac{R_t}{R_{PB}}\right) \Big/ \left(\frac{W_t}{W_{PB}}\right) \qquad (1)$$

The values of $F_t$ were plotted against $t$ to visualize the recovery of fluorescence. For further analysis, the data obtained were fitted to the following exponential function using the Solver tool of Microsoft Excel 2019:

$$f(t) = A \times (1 - e^{-r \times t}) + N \qquad (2)$$

where $r$ is the rate constant, $A$ is the amplitude of the signal change and $N$ the normalized integrated fluorescence intensity after application of

the laser pulse. The recovery half-time ($t_{1/2}$) was then determined using to the equation

$$t_{1/2} = (\ln 2)/r. \qquad (3)$$

### Preparation of samples for super-resolution imaging and single-particle tracking

Cells were picked from freshly plated stocks and grown for ~24 h in 5 ml TS medium, shaking at 600 rpm in sterile tubes. Subsequently, they were diluted to an $OD_{600}$ of 0.02 (or 0.01 for experiments using cefalexin) and grown to the mid-exponential phase ($OD_{600}$ of 0.3-0.5) prior to analysis. Slides were cleaned with 1 M KOH overnight, rinsed with $_{dd}H_2O$, and dried with pressurized air. Cells were imaged on pads composed of 1.5% (w/v) low-melting agarose (agarose, low gelling temperature; Sigma-Aldrich) in sterile-filtered *R. rubrum* minimal medium[73], prepared with the aid of gene frames (Thermo Fisher Scientific).

### Semi-quantitative single-molecule localization microscopy (SMLM)

A sample (1 ml) of a culture was mixed with formaldehyde to a final concentration of 1% and incubated for 30 min at 28 °C and 600 rpm. After centrifugation for 3 min at 4000 × *g* and room temperature, the cells were resuspended in *R. rubrum* minimal medium[73] supplemented with 10 mM glycine and incubated at room temperature in the dark for 5 min to quench residual formaldehyde. This quenching step was repeated three times. In the last round, the suspension was adjusted to a final $OD_{600}$ of ~2.0 prior to incubation. Cells were then imaged by semi-quantitative SMLM with a ZEISS Elyra 7 inverted microscope that was controlled by the ZEN 3.0 SR (black edition) software and equipped with two pco.edge sCMOS 4.2 CL HS cameras (PCO AG), connected through a DuoLink (Zeiss) and an alpha Plan-Apochromat 63×/1.46 Oil Korr M27 Var2 objective in combination with an Optovar 1× (Zeiss) magnification changer (97 nm pixel). During image acquisition, the focus was maintained with a Definite Focus.2 system (Zeiss). To image PapS-PAmCherry and PAmCherry-Por39, a total of 180,000 and 30,000 frames respectively were recorded for each field of view. During each acquisition, 10,000 frames were acquired, using a 561 nm laser (100 mW) set to 50% intensity in TIRF mode (62° angle). To photoactivate PAmCherry, cells were illuminated with a 405 nm laser (50 mW diode laser), whose intensity was increased linearly over the course of the experiment to maintain an approximately constant frequency of PAmCherry photoactivation events. Each frame comprised an integration phase of 50 ms, in which only the 561 nm laser was active, and a transfer phase of ~4 ms, in which only the 405 nm laser was active.

### 3D structured illumination microscopy (3D-SIM)

Cells were harvested by centrifugation for 3 min at 4000 × *g* and 28 °C and resuspended in *R. rubrum* minimal medium[73] to an $OD_{600}$ of ~2.0. Imaging was performed with a ZEISS Elyra 7 with Lattice SIM² microscope equipped with an alpha Plan-Apochromat 63×/1.46 Oil Korr M27 Var2 objective, an Optovar 1.6 × (Zeiss) magnification changer (63 nm pixel) and a Definite Focus.2 system (Zeiss). For dual-color imaging, mCherry and mNeonGreen were imaged sequentially. First, mCherry was excited with a 561 nm laser (100 mW, power intensity: 20%, grating: 27.5 µm G5), and then mNeonGreen was excited with a 488 nm laser (100 mW, power intensity: 20%, grating: 23.0 µm G6). Signals were observed through a multiple beam splitter (405/488/561/641 nm) and laser block filters (405/488/561/641 nm) followed by a Duolink SR DUO (Zeiss) filter module (secondary beam splitter: LP 560, emission filters: BP495-590 + LP570). Dual-color images of FM4-64 and mNeonGreen were obtained by simultaneous imaging of the two fluorophores. To this end, the two fluorophores were excited with a 488 nm laser (100 mW, power intensity: 30%, grating: 27.5 µm G5) and

signals were observed through a multiple beam splitter (405/488/561/641 nm) and laser block filters (405/488/561/641 nm) followed by a DuoLink SR DUO (Zeiss) filter module (secondary beam splitter: LP 640, emission filters: BP495-590 + LP655). In both cases, only 9 phases and 3D leap mode were used (9 Z-slices, 302 nm apart) to reduce bleaching. SIM images were reconstructed using the 3D SIM² module integrated in the ZEN 3.0 SR (black edition) software (Zeiss). Final 3D-SIM snap-shot images were obtained by generating maximum projections of the Z-slices obtained.

### Single-particle tracking (SPT)

Cells were harvested by centrifugation for 3 min at 4,000 × *g* and 28 °C and resuspended in *R. rubrum* minimal medium[73] to an $OD_{600}$ of ~2.0. Imaging was performed with a ZEISS Elyra 7 with Lattice SIM² microscope equipped with an alpha Plan-Apochromat 63×/1.46 Oil Korr M27 Var2 objective, an Optovar 1.6 × (Zeiss) magnification changer (63 nm pixel) and a Definite Focus.2 system (Zeiss). To image PapS-PAmCherry or PAmCherry-Por39, 10,000 frames were recorded, using a 561 nm laser set (100 mW) set to 30% intensity in TIRF mode (62° angle). To photo-activate PAmCherry, the sample was illuminated with a 405 nm diode laser (50 mW), whose intensity was increased linearly from 0 to 0.013% over the course of the experiment. Each frame comprised an integration phase of 20 ms, in which only the 561 nm laser was active, and a transfer phase of ~4 ms, in which only the 405 nm laser was active.

To image mNeonGreen-RodZ, the focal plane of the microscope was set either to the center plane or the peripheral plane of the cells (Fig. 9c). Subsequently, 5000 frames were recorded, using a 488 nm laser (100 mW) at 35% intensity in TIRF mode, with the incidence angle adjusted to 62° (center plane) or 64° (peripheral plane). In both cases, each frame comprised an integration phase of 20 ms, in which the 488 nm laser was active, and a transfer phase of ~4 ms, in which the 488 nm laser was turned off.

To reconstruct particle tracks for PapS-PAmCherry or PAmCherry-Por39, spots were identified with the LoG Detector of Track-Mate v6.0.1[74], as implemented in Fiji 1.53 g[67,75], an estimated diameter of 0.5 µm, and median filter and sub-pixel localization activated. The signal-to-noise threshold for the identification of the spots was set at 5. To limit the detection of ambiguous signal, frames recorded during the bleaching phase (first 2000 frames) were removed from the time-lapse series prior to the identification of spots. Spots were merged into tracks via the Simple LAP Tracker of TrackMate, with a maximum linking distance of 300 nm, one frame gaps allowed, and a gap closing max distance of 500 nm. Only tracks with a minimum length of 5 frames were used for further analysis, yielding a minimum of 711 tracks (Supplementary Data 3). To define the regions of interest (ROIs) for the downstream analysis, the cell outlines were determined with Oufti[76], enabling the exclusion of tracks that originated from background signal outside cells and from regions containing overlapping cells.

The reconstruction of mNG-RodZ tracks was performed in the same manner, but only the first 200 frames were removed from each time-lapse series before spot detection, yielding a minimum of 2109 tracks (Supplementary Data 3). A comparison between tracks localizing at the inner and outer curve, respectively, was performed on the same dataset that was used to determine the global dynamics of mNG-RodZ in the center plane. To this end, new ROIs, one for the inner curve and one for the outer curve, were added to previously analyzed cells with Oufti[76], excluding cells that were not clearly curved (7 out of 216 cells).

To identify differences in protein mobility and/or behavior, the resulting tracks were subjected to mean-squared-displacement (MSD) and squared-displacement (SQD) analysis, as described previously[77,78]. All analytical approaches were used as implemented in SMTracker 2.0[79]. The average MSD was calculated for four separate time points per strain (exposure of 20 ms; τ = 24, 48, 72, and 96 ms), followed by

fitting of the data to a linear equation. The last time point of each track was excluded to avoid track-ending-related artifacts. The cumulative probability distribution of the square displacements (SQDs) was used to estimate the apparent diffusion constants and relative fractions of up to three diffusive states ('fast diffusive, 'diffusive' and 'immobile')[79]. The models were compared using an F-test to determine whether an increase in the number of diffusive states could be justified. More complex models were only accepted if a statistically significant F-test ($p$ value < 0.05) was coupled to a decrease in the associated Bayesian information criterion (BIC) of more than 5%. The apparent diffusion constants of the diffusive and immobile subpopulations were determined globally and then constrained to be equal in the different conditions to enable a more direct comparison of the relative population sizes (see Supplementary Data 3 for information about the strains compared in this manner).

## Cryo-electron microscopy

To determine the width of the periplasmic space of *R. rubrum* cells, cells were grown in TS broth with aeration to an $OD_{600}$ of 0.4-0.5 and then concentrated by centrifugation to an $OD_{600}$ of ~15. Samples (3 µl) of the suspensions were applied to Quantifoil R2/2 holey carbon grids and vitrified in liquid ethane using an FEI vitrobot. Transmission images of vitrified cells were captured using an FEI F20 electron microscope equipped with a Falcon II direct electron detector at an acceleration voltage of 200 kV. Images were taken at a nominal magnification of 11,000×. Periplasm width measurements were made manually from individual micrographs using the line tool in Fiji (version 2.1.0/1.53c)[67,75].

## Immunoblot analysis

Antibodies against PapS were raised by immunization of rabbits with purified PapS-His$_6$ protein (Eurogentec, Belgium). Cells were harvested at an $OD_{600}$ of ~0.4 and subjected to immunoblot analysis as described previously[80], using anti-PapS antiserum, a monoclonal anti-mNeonGreen antibody (Chromotek, Germany; Cat. #: 32f6; RRID: AB_2827566), a monoclonal anti-mNeonGreen antibody (Chromotek, Germany; Cat. #: 55074) or a polyclonal anti-mCherry antibody (Bio-Vision, USA; Cat. #: 5993-100) at dilutions of 1:30,000, 1:1,000, 1:1,000 or 1:10,000, respectively. PageRuler™ Plus Prestained Protein Ladder (Thermo Fisher Scientific, USA) was used as a protein standard. Goat anti-rabbit immunoglobulin G conjugated with horseradish peroxidase (Revvity, USA; Cat. #: NEF812E001EA; at a 1:20,000 dilution) was used as a secondary antibody. Immunocomplexes were detected with the Western Lightning Plus-ECL chemiluminescence reagent (Perkin Elmer, USA). The signals were recorded with a ChemiDoc MP imaging system (BioRad, Germany) and analyzed using Image Lab software (BioRad, Germany).

## Isolation of peptidoglycan sacculi

For imaging, sacculi of *R. rubrum* S1 and JR52 were purified as described previously[81] with minor adjustments. In short, 200 ml cultures ($OD_{600}$ of ~0.4) were harvested by centrifugation for 10 min at 6340 × $g$ and 4 °C. The cells were washed once in 5 ml of buffer S1 (10 mM Tris/HCl, pH 6.8), harvested again and resuspended in 2 ml of buffer S1. The suspension was then added dropwise to 15 ml of boiling 6% SDS. Subsequently, the solution was kept boiling for 45 min, with its volume kept constant by the addition of double-deionized water. The mixture was then cooled to room temperature, and sacculi were harvested at 200,000 × $g$ for 30 min at 20 °C. For storage, sacculi were resuspended in 1 ml double-deionized water, snap-frozen in liquid nitrogen and stored at -80 °C.

## Muropeptide analysis

Exponentially growing cells (250 ml) were rapidly cooled to 4 °C and harvested by centrifugation at 16,000 × $g$ for 30 min. Subsequently,

they were resuspended in 6 ml of ice-cold $H_2O$ and added dropwise to 6 ml of a boiling solution of 8% sodium dodecylsulfate (SDS) that was stirred vigorously. After 30 min of boiling, the suspension was cooled to room temperature. Peptidoglycan was isolated as described previously[82] and digested with the muramidase cellosyl (kindly provided by Hoechst, Frankfurt, Germany). The resulting muropeptides were reduced with sodium borohydride and separated by HPLC following an established protocol[82,83], except that 30% methanol was used in elution buffer B. The identity of eluted fragments was assigned based on the retention times of known muropeptides[84].

## Triton extraction of porin-PapS complexes

To extract the porin-PapS complex, 30 ml of an *R. rubrum* culture (OD of ~0.5) were harvested by centrifugation for 10 min at 5400 × $g$ and 4 °C. The cells were washed twice in 10 ml of ZAP buffer (20 mM Tris/HCl, 10 mM MgCl$_2$, pH 8) containing 250 µM phenylmethylsulfonyl fluoride (PMSF) and lysed by three passages through a French press at 16,000 psi. Insoluble cell fragments were collected by centrifugation for 10 min at 20,000 × $g$ and 4 °C, washed once in 2 ml of ZAP buffer and then incubated in 2 ml of Triton extraction buffer (20 mM Tris/HCl, 10 mM MgCl$_2$, 250 µM PMSF, 2% Triton, pH 8) for 20 min at 40 °C and 1400 rpm in a Thermomixer Comfort (Eppendorf, Germany). Residual cell debris was removed by centrifugation for 5 min at 20,000 × $g$ and room temperature. Subsequently, the supernatant was dialyzed against ZAP buffer and then tenfold diluted ZAP buffer overnight at 4 °C and concentrated to a final volume of 500 µl using Amicon Ultra-4 centrifugal filters (Millipore, USA).

## Total proteome analysis using shotgun proteomics and liquid chromatography-mass spectrometry

$3 \times 10^9$ cells from an exponentially growing culture were pelleted by centrifugation, resuspended in 300 µl of lysis buffer (100 mM ammonium bicarbonate, 2% sodium laroyl sarcosinate [SLS]), and lysed by incubation for 5 min at 95 °C and subsequent ultra-sonication for 10 sec (Vial Tweeter, Hielscher). The cell lysate was incubated for 15 min with 5 mM Tris(2-carboxyethyl)phosphine (TCEP) at 90 °C followed by alkylation with 10 mM iodoacetamide for 30 min at 25 °C. After the removal of cell debris by centrifugation for 10 min at 21,230 × $g$, the supernatant was transferred into a new tube and its protein content was quantified using a BCA Protein Assay kit (Thermo Fisher Scientific). SLS was diluted to 0.5 % and 50 µg protein was digested by the addition of 1 µg trypsin (Serva) followed by incubation at 30 °C overnight. The solution was mixed with trifluoroacetic acid (TFA) to a final concentration of 1.5 % and incubated at room temperature for 10 min to precipitate SLS. After centrifugation for 10 min at 9400 × $g$, peptides were purified from the supernatant using Chromabond C18 SpinColumns (Macherey-Nagel), dried, and resuspended in 0.1 % TFA. The concentration of peptides in the samples was measured with a colorimetric assay (Pierce™ Quantitative Colorimetric Peptide Assay, Thermo Fischer Scientific).

The peptides were analyzed with a Q-Exactive Plus mass spectrometer coupled to an Ultimate 3000 RSLC nano with a Prowflow upgrade and a nanospray flex ion source (Thermo Scientific). They were separated on a reversed-phase HPLC column (75 µm x 42 cm) packed in-house with C18 resin (2.4 µm, Dr. Maisch GmbH, Germany) using the following gradient at a flow rate of 300 nl min$^{-1}$: 94% solvent A (0.15% formic acid) and 6% solvent B (99.85 % acetonitrile, 0.15 % formic acid) to 20% solvent B over 40 min, and to 35% solvent B over another 20 min. The mass spectrometric analysis was performed as described previously[85].

Label-free quantification (LFQ) of the data was performed with MaxQuant[86] in standard settings with the "match between runs" option, using a UniProt protein FASTA database of *Rhodospirillum rubrum* (strain ATCC 11170). The search criteria were set as follows: full tryptic specificity was required (cleavage after lysine or arginine

residues); two missed cleavages were allowed; carbamidomethylation (C) was set as a fixed modification; oxidation (M) and deamidation (N,Q) were set as variable modifications. Protein iBAQ values were calculated within MaxQuant. A detailed list of all MaxQuant parameters is provided in Supplementary Data 1.

## Protein purification

His$_6$-SUMO-tagged[87] PapS was overproduced in *E. coli* SHuffle T7 (New England Biolabs, USA) transformed with pJR87, whereas His$_6$-SUMO-tagged OmpA-like domains were overproduced in *E. coli* Rosetta(DE3) pLysS (Thermo Fisher Scientific, USA) bearing pSP74, pSP78 or pSP213. Cultures were grown in LB medium at 37 °C to an OD$_{600}$ of 0.8. Subsequently, protein overproduction was induced by the addition isopropyl-β-D-thiogalactopyranoside (IPTG) to a final concentration of 0.5 mM, and cultivation was continued for 3 h at 37 °C or overnight at 18 °C. The cells were harvested by centrifugation, washed with buffer BZ3 (50 mM Tris, 300 mM NaCl, 10% v/v glycerol, 20 mM imidazole, adjusted to pH 8 with HCl), and stored at −80 °C until further use. After thawing, they were resuspended in buffer BZ3 (2 ml per 1 g of pellet) supplemented with DNase I (10 µg ml$^{-1}$) and phenylmethylsulfonyl fluoride (100 µg ml$^{-1}$). After three passages through a French press at 16,000 psi, cell debris was removed by centrifugation at 30,000 × *g* for 30 min. The supernatant was then applied onto a 5 ml HisTrap HP column (GE Healthcare, USA) equilibrated with buffer BZ3. The column was washed with 5 column volumes (CV) of buffer BZ3, and the protein was eluted with a linear imidazole gradient (20–250 mM in buffer BZ3) at a flow rate of 1 ml min$^{-1}$. Fractions containing the protein at high purity were pooled, supplemented with 1 mM DTT and His$_6$-Ulp1[87] protease at a 1:1000 molar ratio relative to the His$_6$-SUMO-tagged protein of interest and dialyzed against buffer CB (50 mM Tris-HCl, 150 mM KCl, 10% glycerol, pH 8) overnight. Subsequently, the solution was applied onto a 5 ml HisTrap HP column (GE Healthcare, USA) equilibrated with buffer CB as described above. Flow-through fractions containing PapS at high purity were pooled, aliquoted, snap-frozen, and stored at −80 °C.

## Size-exclusion chromatography

Purified PapS (1 mg ml$^{-1}$) was loaded onto a Superdex 75 10/300 GL Precision Column (GE Healthcare) that was equilibrated with CB buffer (50 mM Tris-HCl, 150 mM KCl, 10% glycerol, pH 8). Protein was eluted at a flow rate of 0.3 ml min$^{-1}$. A mixture of standard proteins, including 0.7 mg ml$^{-1}$ Ribonuclease A (13.7 kDa), 0.7 mg ml$^{-1}$ Ovalbumin (43 kDa), and 0.7 mg ml$^{-1}$ Conalbumin (76 kDa), was analyzed as a reference.

## Microscale thermophoresis

Meso-diaminopimelic acid (mDAP)-binding assays were performed with a Monolith NT.115 instrument controlled by the NTControl v2.1.33 software, using Monolith NT Premium Capillaries or Monolith NT.155 Standard Treated Capillaries (NanoTemper Technologies GmbH, Germany). Purified proteins were transferred into MST buffer (25 mM HEPES/NaOH pH 8, 50 mM NaCl, 0.1% Tween) using the PD SpinTrapTM G-25 kit (cytiva, USA). Proteins were fluorescently labeled using the RED-NHS 2$^{nd}$ Generation Protein Labeling Kit (NanoTemper Technologies GmbH, Germany) as recommended by the manufacturer. For titrations, 100 nM labeled protein was analyzed in the presence of mDAP at concentrations ranging from 1.5 µM to 50 mM for the peptidoglycan-binding domain of PapS and from 61 nM to 2 mM for the peptideglycan-binding domain of *A. baumannii* OmpA. Measurements were performed at 25 °C, with the red LED laser adjusted to a power of 90% and the infrared laser set to 60%. Three to four independent measurement (three technical replicates each) were performed for each condition. Data were analyzed using MO Affinity Analysis v2.3 (NanoTemper Technologies GmbH, Germany). To prevent heat-induced artifacts and, at the same time, avoid analyzing only

the signal change induced by the initial temperature jump, the following regions were used for data analysis: cold region from −1 s to 0 s, hot region from 1.5 s to 2.5 s.

## Bioinformatical analysis

Protein similarity searches were performed with BLAST[88], using the BLAST server of the National Institutes of Health (https://blast.ncbi.nlm.nih.gov/Blast.cgi). Hits with e-values of 10$^{-15}$ or lower were considered to be significant. phyloT v2 (https://phylot.biobyte.de) was used for phylogenetic tree construction, while iTol 6.5.2[38] was used for tree visualization. MAFFT 7[38] was used for protein sequence alignments. WebLogo 3[89] was used for the generation of sequence alignment logos. Conserved protein domains were determined using the PFAM server[90]. SignalP 6.0[91] was used to identify signal peptides with default gathering thresholds. AlphaFold2[38] and AlphaFold-Multimer[48] were used for protein structure prediction. The structural coordinates and error estimates for the models obtained are provided in Supplementary Data 7. Protein structures were visualized using PyMOL (version 2.6.0a0; Schrödinger, LLC). The Super-PlotsOfData web app[69] was used to visualize the distribution of data and evaluate the statistical significance of differences between multiple distributions.

## Statistics and reproducibility

All experiments were performed at least twice independently with similar results. No data were excluded from the analyses. Calculations and statistical analyses were performed in Excel 2019 (Microsoft), unless indicated otherwise. The statistical significance of differences between datasets was determined using unpaired two-sided Welch's *t*-tests. Protein localization patterns were analyzed at least three times independently with similar results. To quantify imaging data, multiple images were analyzed per condition. The analyses included all cells in the images or, in the case of high cell densities, all cells in a square portion of the images. The selection of the images and fields of cells analyzed was performed randomly.

## Availability of biological material

The plasmids and strains used in this study are available from the corresponding author upon request.

## Reporting summary

Further information on research design is available in the Nature Portfolio Reporting Summary linked to this article.

## Data availability

The mass spectrometry proteomics data generated in this study have been deposited to the ProteomeXchange Consortium via the PRIDE partner repository with the dataset identifier PXD049473. All other data generated in this study are included in the manuscript or the supplementary material. Source data are provided with this paper.

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

## Acknowledgements

We thank Lara Stuhl for help with the construction of strains, Juri Hanßmann for advice on the microscale thermophoresis analysis,

Daniela Vollmer for the purification of peptidoglycan, Seán Murray for helpful discussions, Morgan Beeby for support with the cryo-electron tomography analysis and Jürgen Weckesser for supporting initial experiments on PapS function. This work was funded by the University of Marburg (core funding to M.T.), the Max Planck Society (Core funding to T.G. and Max Planck Fellowship to M.T.), the German Research Foundation (DFG; project 269423233 – TRR 174 to M.B. and M.T., project 450420164 – TH 885/3-1 to M.T. and project INST 257/645-1 to M.B.), and the United Kingdom Biotechnology and Biological Sciences Research Council (grant BB/W013630/1 to W.V.).

## Author contributions

S.P., M.C.F.v.T and M.T. conceived the study. S.P. constructed plasmids and strains and performed the widefield imaging-based localization studies, sinuosity measurements, growth analyses and biochemical experiments. G.G. performed the SMLM, 3D-SIM, SPT and TIRF analyses. F.M.M. performed the outer-versus-inner-curve signal ratio quantification. V.K. and J.H. contributed initial gene-disruption analyses suggesting a role of *papS* in cell morphogenesis. E.J.C. performed the cryo-EM analysis and the periplasmic width measurements. J.B. conducted the peptidoglycan analysis. J.R. constructed plasmids and strains. T.G. performed the mass spectrometry analysis. S.P., G.G., F.M.M., E.J.C., J.B., T.G., W.V., J.H. and M.T. analyzed the data. W.V., M.C.F.v.T, J.H., M.B. and M.T. supervised the study. W.V., M.B. and M.T. (with support by M.C.F.v.T) secured funding. S.P. and M.T. wrote the paper, with input from all other authors.

## Funding

## Competing interests

The authors declare no competing interests.
