## [Peer Review File · Nature Communications]

An outer membrane porin-lipoprotein complex modulates
elongasome movement to establish cell curvature in
Rhodospirillum rubrumReviewer #1 (Remarks to the Author):

I read with a great interest this very interesting study, in which the authors demonstrate that the conserved membrane porin-lipoprotein complex Por39-41/PapS governs the establishment of cell curvature in *Rhodospirillum rubrum*. This is a very well-written manuscript that describes a series of carefully performed experiments, based on a remarkable array of genetic, biochemical, structural modeling and microscopic approaches. The conclusions are fully supported by the experiments made and it represents a very nice contribution to the understanding of morphogenetic mechanisms of the bacterial cell, and notably those for bacteria with a curved shape. It leaves little room for criticism. I think the results presented here will be of great interest to microbiologists. I have no objections and I think this work can be published as it is. A minor comment would be to add in Figure 10 a molecular model of the complex and its organization in the membrane. The author may use it to comment more on the differential contribution of Por39 and 41, notably regarding the observations made with the mutations of the conserved D71 and P282(292) and T280(V290). They may also use it to speculate a bit on the importance of their oligomerization mode (homo-, hetero-trimers) in the interplay with PapS. Two other points, just out of curiosity: did the authors analyze the dynamic of MreB in their PapS mutants? If yes, is it different from that of RodZ? Do the authors envision an interplay between PapS and a specific component of the elongasome? Direct or do they think that a factor is still missing?

Reviewer #2 (Remarks to the Author):

In this manuscript, Poehl et al. perform a characterization of the PapS protein, which is required for the curved morphology of *Rhodospirillum rubrum*. PapS localizes to the periplasm, and mutations abolishing its ability to bind to the outer membrane or to peptidoglycan abolish its cell-curving effect. PapS stably localizes to the outer curvature of curved cells, and its localization is dependent on its ability to bind the OM porins Por39 and 41, with which it was previously found to form stable complexes and which also localize to the outer cell curvatures. Evidence is presented that PapS enriches elongasomes (using RodZ as a proxy) at the outer curvature; these outer curvature-localized elongasomes display reduced mobility (relative to Δ papS), which indeed may lead to their enrichment. These data collectively lead the authors to propose a model in which PapS-mediated enrichment and mobility reduction of elongasomes along one side of the cell lead to an asymmetry in peptidoglycan insertion along the long axis of the cell and hence cell curvature.

This is a beautifully conducted and presented study. The most important and interesting finding of this work is that it reveals yet another heretofore unknown, mechanistically distinct way in which bacteria can achieve a curved morphology. The authors are very thorough, using a variety of methods to rule out possibilities (many such data are in the 16-figure supplement) and to support an appealing model. Overall, the conclusions of the study are very well-supported by the experiments as presented. There remain only a few relatively minor questions and concerns.

Principal questions and concerns

1. The model in ll. 370-375 and Fig. 10 is appealing and in no way contradicted by the data. Certainly RodZ is enriched at the outer curvature. But the data in Fig. 9, while comparing single-particle RodZ mobility between WT and Δ papS cells at the center and peripheral planes, does not compare its mobility between the inner and outer curvatures of the cells. The data are likely already present in existing datasets (cf. Fig. S15). A difference in mobility between the two sides would strengthen the "roadblock" model.
2. The authors should comment on the additional protein bands seen for FP- or Bla-tagged versions of PapS that are not seen in the WT (Figs. S1b, S3b, S5b, S7c, S10e, S11b, S14c). Are these cleavage products? Do the authors think that this distinct banding pattern is related to the reduced sinuosity of strains with C-terminal PapS fusions (Figs. 3c, S1a, S3a, S5c, S6b, S7e, S9b, S10c, f) relative to strains with wild-type PapS?
3. The interpretation in ll. 386-389 combining the present work with recent work from the same group to posit that BacA antagonizes PapS-mediated curvature by stimulating PG synthesis at the

inner curve is quite interesting. However, the observation that Δ papS cells are straight (Fig. 1) suggests that BacA cannot asymmetrically bias cell elongation in Δ papS cells, either because its localization depends on PapS or on cell curvature itself. Hence, it would be interesting to examine BacA-mCh localization in a Δ papS background to learn if it becomes delocalized, thus hierarchically linking these two cell curvature control systems.

4. I do not regard the following experiments as critical for this manuscript, but they would nonetheless be interesting with respect to the conservation of PapS-mediated curvature. The proposal that the cell-curving activity of PapS is widely conserved (II. 108-9) could be experimentally validated by deleting papS in at least one other representative of the Rhodospirillales. Additionally, the general ability of PapS to induce curvature could be tested in one or more straight species. While it might not work in, say, *E. coli* due to different OM proteins, it might work in a straight species more closely related to *R. rubrum*.

Very minor issues

II. 264-65, should be PapS-mNG to match the figure.

In immunoblots in the supplement, arrowheads are frequently used but only described in the legend of Fig. S5.

Fig. S11 uses asterisks for significance estimates that are not defined in the legend; the other figures show the p-value directly.

Reviewer #3 (Remarks to the Author):

In this manuscript Pöhl et al. set out to describe a cell biological mechanism that the bacterium *Rhodospirillum rubrum* uses to generate its characteristic helical rod morphology. This manuscript has an impressive number of discoveries and uses a variety of experimental approaches including advanced fluorescence microscopy and structural biology approaches. In general, the authors should be congratulated for preparing an extensive manuscript clearly describing the role of (newly-named) PapS and interactions with Por41, Por39, and the elongasome. In addition to further details about their individual claims described below, there are two major areas that need to be resolved. In brief, (1) while the authors predict an ~15% rate of outer curve extension relative to inner curve extension, they do not see this in their HADA labeling. This could be because of a distinction between the *rate* of extension/expansion/growth as compared to the *amount* of growth. Indeed, if there is a difference in the local rate of expansion, one would expect shape *changes* as opposed to the observed shape *maintenance*. (2) the single particle tracking analysis was somewhat confusing and needs to be described further, as well as described more fully in the context of other elongasome tracking data. The role of an outer membrane complex to feed back on shape is incredibly fascinating, and represents a novel strategy that bacteria can utilize to pattern the geometric growth of their wall, thereby patterning their stereotypical shape to grow and thrive.

Given the length of this manuscript, this reviewer's comments have been organized into three major claims/discoveries made in this manuscript, and a follow-up section with other minor points and style suggestions. Except where noted, the description of the methods, approaches, and analysis are well described with sufficient detail.

Claim 1: There is an outer-membrane protein complex in *R. rubrum* that promotes cell curvature, and this complex includes the lipoprotein PapS and porins Por39 and Por41. The support for claim 1 is very strong, including genetic disruption of PapS resulting in straight cells and straight sacculi. These shapes can be rescued by expression of PapS from a plasmid. The authors do a great job explaining their interpretation that the conservation of PapS in other curved Rhodospirillales implies PapS function is not limited to *R. rubrum*.

Comment 1.1: Strengthening the claim to prove that PapS is used in other species would likely require genetic tools and culturability in those species, and is likely beyond the scope of this study.

Claim 2: Porins Por39 and Por41 and PapS form helical ribbon-like structures at the outer curve of

the cell, with the localization of PapS mediated by the porins. The authors generate semi-functional fusions of PapS fused to few different fluorescent proteins, all of which restore some cell curvature. Modifications predicted to disrupt the membrane binding and peptidoglycan binding domains showed that peptidoglycan binding was required for mediating cell curvature and outer membrane binding was also somewhat important. Following up with these PapS domain disruptions, the authors also examined PapS localization in the presence of Por39/Por41 disruptions. Unfortunately, entirely deleting these porins was not possible, so the authors utilized point mutants to dissect the order of assembly and localization, concluding that Por41 has the dominant role in positioning PapS assemblies. Finally, the authors dissect the binding interfaces on Por39/Por41/PapS for the complex and are able to generate point mutants that disrupt the interface. Overall, the support for claim 2 is strong.

Comment 2.1: line 158 claims protein accumulation was less than in WT-fusion, however none of the Western blots show a loading control to substantiate any claims of the relative amount of protein in any lane. Ideally, these loading controls would be included in all blots, but in the absence of this, claims about the abundance of protein from one lane to another should be removed.

Comment 2.2: The authors claim these putative disruptions remove membrane binding and peptidoglycan binding but do not provide evidence that they do. For example, is the purified PapS able to bind purified sacculi but the PG binding deficient one is not?

Comment 2.3: There should be some discussion and/or acknowledgement that the fluorescent fusions do not restore cell curvature completely, and restore it to different amounts in different constructs. This is to be expected in fluorescent fusions, and should be discussed somewhere.

Comment 2.4: There are a multiple prominent bands in S3B, S5B that run at different weights. There should be some, even minor, discussion of this because it implies to the reader that there are different complexes or protein abundances in these strains. In S3B, are these breakdown products? are those products fluorescent? How does this influence the interpretation of the fluorescent localization?

Minor Comment 2.5: The brief mention of FM4-64 is not as clearly defined as many of the other observations. Does this localization of FM4-64 depend on the Por39/Por41/PapS complex? In straight(er) cells that still have PapS with its membrane binding domain intact, is there preferential lipid stain? In straight cells lacking PapS entirely, does one face of the cell still have an FM4-64 enrichment?

Claim 3: These porin complexes trap the elongation machinery, leading to biased growth such that the outer curve grows faster than the inner curve, leading to overall cell curvature. The authors use a combination of tracking wall synthesis with fluorescent D-amino acids and single-particle tracking of various complexes to conclude that the elongosome components are trapped and spend more time on the outer curve. In addition to this HADA labeling, the authors went on to examine the localization and dynamics of a component of the elongosome, RodZ. This claim has the weakest support overall. In light of the conceptual issues described here, it is difficult to assess the author's overall model that porin complexes trap the elongosome.

Major comment 3.1: The authors comment that they expect a 15% increase in HADA staining on the outer curve, but do not observe any enrichment. This conclusion is complicated by the fact that *R. rubrum* maintains its relative curvature throughout the growth phase. While the outer curve may be 15% longer, it is not immediately obvious that the outer curve has to grow faster per unit surface area. This is a major point that needs to be resolved in the language used to describe the dynamics of shape establishment as compared to shape propagation. This also applies to the language and description in the overall working model of Figure 10. The author do not demonstrate evidence of going from a straight rod to a helical rod, rather disrupted shapes that happily propagate with altered geometry. This same mechanism may be relevant for establishment of shape, but here the authors simply test shape maintenance. In regards to deciphering the role of PapS/Por39/Por41 in maintaining shape, the authors do an excellent job and should be commended on their logical progression and description of the system.

Major comment 3.2: The single particle tracking analysis throughout was confusing. For many previously published studies of single particle tracking of elongasome components, trajectories were either measured in just the peripheral plane [<https://www.nature.com/articles/ncomms13170>, <https://elifesciences.org/articles/32471>] or in an unwrapped 3D surface of the cell [<https://www.pnas.org/doi/full/10.1073/pnas.1509610112>]. The authors need to include a more careful description of their analysis. The bubble plots for mixed populations of dynamics appear to utilize a global definition of D1 and D2, and allow only a fitting parameter for the fraction of each population. However, the influence of cell shape, and sub-cellular compartmentalization on trajectories is not discussed in the current manuscript. See [<https://www.sciencedirect.com/science/article/abs/pii/S0378437106012039>, <https://elifesciences.org/articles/82654>, <https://www.pnas.org/doi/full/10.1073/pnas.1102255108>]. The relative mobility of the elongasome complexes is confusing and should be more carefully compared to previous literature values. For example, is the slowly moving complex with a $D_{eff} = 0.0191 \text{ } \mu\text{m}^2 \text{ s}^{-1}$ truly moving? Or does this more closely match the faster two subpopulations observed in <https://elifesciences.org/articles/50629>? As a followup to needing more details about these procedures, the methods section describes the illumination for the tracking as being in "TIRF mode", which would imply the illumination intensity as limited to the peripheral plane and not illuminate anything in the center plane.

In addition to comments and suggestions about these specific claims, here are a few minor stylistic changes that will improve the readability overall.

Minor comment 4.1: In addition to the sinuosity of delta PapS cells in Supp 1A, it would have been nice to see a delta PapS with an empty vector control in Supp 3A to ensure that plasmid bearing strains are straight unless they have PapS. It also appears as if the data for the WT cells is replotted/identical in Supp 1A and Supp 3A, which would be nice to include as a comment in the figure captions.

Minor comment 4.2: In supplemental figure 1E, recommend removing bar graph and using just the simple colored dots to represent each experiment. [<https://github.com/cxli233/FriendsDontLetFriends>]

Minor comment 4.3: line 55, Por41 and P39 => should be Por41 and Por39?

Minor comment 4.4: Figure S3CD, I was expecting representative phase images and quantification of the fluorescent strains as well.

Minor comment 4.5: Figure 3C, S5C, pValues are very confusing in the way they are presented, maybe add some heads to the lines, or make sure they line up appropriately? Also, if using scientific notation on some pValues, please use it on all.

Minor comment 4.6: This reviewer experienced confusion in the characterization purified PapS and the differences between different PapS fusions. In general the authors do a good job clarifying the difference between fluorescent fusions and untagged proteins, but only include size exclusion chromatography for the non-fusion PapS construct.

Minor comment 4.7: To help the reader contextualize the scale for sinuosity, maybe there could be a supplementary figure with a few toy examples of different sinuosity, or phase contrast images of individual cells as real examples that show a range from 1 to 1.05 or even 1.08.

Minor comment 4.8: Please check for order of subfigures, it would be easier for the reader to have the subfigures remain in their current positions, and have labels that go in order, even if their reference in the main text does not. For example, Figure 4 has labels go a, b, c, d down the left, then jump up to e before returning next to d for f then g and h go left to right and end with i at the top. This is hard to follow. Also applies to other figures, such as Fig 9.

Minor comment 4.9: The label under the columns in S6AB was confusing. The combination of tilted names, and some underlined, and some not underlined was difficult to follow. Maybe the authors could try a different labeling strategy to clarify. This also applies to S5C, S7CDE, S11C although those ended up being less confusing for this reviewer to follow.

Minor comment 4.10: There is a small typo in the legend of figure 6. Subplot 'g' should be 'f' instead.

Reviewer #4 (Remarks to the Author):

Response to reviewers

We thank all four reviewers for their positive feedback and their constructive criticism, which helped to significantly improve our manuscript.

Please see below for a detailed explanation of newly added results and for our response to the issues raised in the first round of review.

Reviewer #1

I read with a great interest this very interesting study, in which the authors demonstrate that the conserved membrane porin-lipoprotein complex Por39-41/PapS governs the establishment of cell curvature in *Rhodospirillum rubrum*. This is a very well-written manuscript that describes a series of carefully performed experiments, based on a remarkable array of genetic, biochemical, structural modeling and microscopic approaches. The conclusions are fully supported by the experiments made and it represents a very nice contribution to the understanding of morphogenetic mechanisms of the bacterial cell, and notably those for bacteria with a curved shape. It leaves little room for criticism. I think the results presented here will be of great interest to microbiologists. I have no objections and I think this work can be published as it is.

A minor comment would be to add in Figure 10 a molecular model of the complex and its organization in the membrane. The author may use it to comment more on the differential contribution of Por39 and 41, notably regarding the observations made with the mutations of the conserved D71 and P282(292) and T280(V290). They may also use it to speculate a bit on the importance of their oligomerization mode (homo-, hetero-trimers) in the interplay with PapS.

We would prefer not to show a model of the complex that goes beyond the AlphaFold2 models shown in **Supplementary Figure 7**, because we do currently not have information about the precise mode of porin oligomerization and its role in modulating the interaction with PapS. Our proteome analysis shows that Por41 is considerably more abundant than Por39, suggesting that the majority of porin complexes may be Por41 homotrimers. This may explain why the mutation of Por39 did not have any major effect on helical organization of the porin-PapS complexes.

To provide more insights into the role of the amino acid residues exchanged in this study, we have now added an additional panel to **Supplementary Figure 7** that highlights their positions in a putative Por39-Por41₂ heterotrimer. It appears that D71 is located close to the inter-subunit interfaces of the porins, which may potentially explain its importance for porin assembly and localization.

Two other points, just or curiosity: did the authors analyze the dynamic of MreB in their PapS mutants? If yes, is it different from that of RodZ?

We tried intensively to generate fluorescently tagged variants of additional elongasome components, including MreB. However, the fusion proteins were either completely non-functional, leading to cell lysis, or not sufficiently functional to maintain cell shape, causing the rounding of cells or a complete loss of the cell curvature. Therefore, our study focused on the fully functional mNG-RodZ fusion protein as a proxy of the elongasome complex.

Do the authors envision an interplay between PapS and a specific component of the elongasome? Direct or do they think that a factor is still missing?

In our co-immunoprecipitation analyses, we did not identify any elongasome components that interact with PapS. We propose that porin-PapS complexes act as roadblocks that sterically hinder the circumferential movement of elongasome complexes, so that the interplay between PapS and the

elongasome is limited to short collisions that do not involve specific protein-protein interactions. However, we cannot fully exclude the existence of additional, so-far unknown factors that are involved in this process.

Reviewer #2

In this manuscript, Poehl et al. perform a characterization of the PapS protein, which is required for the curved morphology of *Rhodospirillum rubrum*. PapS localizes to the periplasm, and mutations abolishing its ability to bind to the outer membrane or to peptidoglycan abolish its cell-curving effect. PapS stably localizes to the outer curvature of curved cells, and its localization is dependent on its ability to bind the OM porins Por39 and 41, with which it was previously found to form stable complexes and which also localize to the outer cell curvatures. Evidence is presented that PapS enriches elongasomes (using RodZ as a proxy) at the outer curvature; these outer curvature-localized elongasomes display reduced mobility (relative to $\Delta papS$), which indeed may lead to their enrichment. These data collectively lead the authors to propose a model in which PapS-mediated enrichment and mobility reduction of elongasomes along one side of the cell lead to an asymmetry in peptidoglycan insertion along the long axis of the cell and hence cell curvature.

This is a beautifully conducted and presented study. The most important and interesting finding of this work is that it reveals yet another heretofore unknown, mechanistically distinct way in which bacteria can achieve a curved morphology. The authors are very thorough, using a variety of methods to rule out possibilities (many such data are in the 16-figure supplement) and to support an appealing model. Overall, the conclusions of the study are very well-supported by the experiments as presented. There remain only a few relatively minor questions and concerns.

Principal questions and concerns

1. The model in ll. 370-375 and Fig. 10 is appealing and in no way contradicted by the data. Certainly RodZ is enriched at the outer curvature. But the data in Fig. 9, while comparing single-particle RodZ mobility between WT and $\Delta papS$ cells at the center and peripheral planes, does not compare its mobility between the inner and outer curvatures of the cells. The data are likely already present in existing datasets (cf. Fig. S15). A difference in mobility between the two sides would strengthen the “roadblock” model.

As suggested by Reviewer #2, we made use of existing data to compare the mobility of mNG-RodZ at the inner and outer curve. The analysis was performed using single-particle tracks that were recorded in the center plane, because the inner and outer curves are not as well-defined for cells imaged in the peripheral plane. It focused exclusively on the WT strain, since $\Delta papS$ cells are not curved. For the SQD analysis, simultaneous two-component fitting was performed for the data collected at the inner curve, the outer curve, and the entirety of the cell. The results obtained demonstrate that [1] the outer curve is enriched in mNG-RodZ tracks and [2] the mobility of RodZ is lower at the outer curve (**Figure 9f,g, Supplementary Figure 15c and Supplementary Data 4**).

Together, these findings strongly support the roadblock model presented in **Figure 10**. The manuscript has been updated to include the new findings.

2. The authors should comment on the additional protein bands seen for FP- or Bla-tagged versions of PapS that are not seen in the WT (Figs. S1b, S3b, S5b, S7c, S10e, S11b, S14c). Are these cleavage products? Do the authors think that this distinct banding pattern is related to the reduced sinuosity of strains with C-terminal PapS fusions (Figs. 3c, S1a, S3a, S5c, S6b, S7e, S9b, S10c, f) relative to strains with wild-type PapS?

Fluorescent proteins show a high resistance against proteolytic degradation. When fused to a protein of interest, this property can lead to the stabilization of proteolytic intermediates that are otherwise rapidly degraded. It is unlikely that the presence of these metastable intermediates affects the function of the PapS system, because they are presumably truncated at the N-terminal end and thus lack the

residues mediating the interaction of PapS with the porins (W22 and W58) and are therefore no longer recruited to the porin-PapS assemblies. The reduced function of the fusion proteins may therefore rather be the result of steric hindrance by the β -lactamase or fluorescent protein tags. We now briefly discuss this issue in the Results section when introducing the PapS-mCherry fusion protein and state in the legend to Figure 1b that the bands of lower molecular weight are degradation products.

The accumulation of fluorescently tagged degradation products could potentially lead to an elevated background signal originating from delocalized fusion proteins. However, the different fluorescently tagged PapS variants all give very clear signals and produce the same localization patterns. Moreover, the patterns observed for the PapS-mNG/mCh-PAMCh fusion proteins are identical to those obtained for mCh-Por39, which hardly shows any degradation products. Together, these observations indicate that the presence of degradation products does affect the quality of the data obtained.

3. The interpretation in ll. 386-389 combining the present work with recent work from the same group to posit that BacA antagonizes PapS-mediated curvature by stimulating PG synthesis at the inner curve is quite interesting. However, the observation that $\Delta papS$ cells are straight (Fig. 1) suggests that BacA cannot asymmetrically bias cell elongation in $\Delta papS$ cells, either because its localization depends on PapS or on cell curvature itself. Hence, it would be interesting to examine BacA-mCh localization in a $\Delta papS$ background to learn if it becomes delocalized, thus hierarchically linking these two cell curvature control systems.

We already performed the proposed experiment before the submission of the present paper. Our results show that the asymmetric localization of BacA and LmdC to the inner curve and their ability to modulate cell shape requires the prior establishment of cell curvature by the PapS system. Based on these findings, we hypothesize that BacA has an intrinsic preference to accumulate on positively curved membranes. The loss of cell curvature in the $\Delta papS$ background thus causes the loss of its native localization pattern, explaining the essential role of PapS in the function of the BacA/LmdC system. The PapS system thus appears to represent the primary cell shape determinant, while the BacA/LmdC plays a subordinate role and only modulates the curvature pre-established by the porin-PapS assemblies. We believe that a detailed report of the interplay between the two systems is beyond the scope of the present study and would therefore prefer to include these results in a follow-up paper.

4. I do not regard the following experiments as critical for this manuscript, but they would nonetheless be interesting with respect to the conservation of PapS-mediated curvature. The proposal that the cell-curving activity of PapS is widely conserved (ll. 108-9) could be experimentally validated by deleting *papS* in at least one other representative of the Rhodospirillales. Additionally, the general ability of PapS to induce curvature could be tested in one or more straight species. While it might not work in, say, *E. coli* due to different OM proteins, it might work in a straight species more closely related to *R. rubrum*.

We agree that it would be interesting to investigate the PapS system in other members of the Rhodospirillales and thus clarify its degree of conservation. However, for the vast majority of these bacteria, genetic tools or fully closed genome sequences are not available. We are currently working on the analysis of PapS in *Magnetospirillum gryphiswaldense*, a species that has been shown to be genetically amenable. However, the genetic modification of this slow-growing, microaerophilic organism is difficult and time-consuming, so that it is not possible to produce results within the time available to revise the current manuscript.

To test the activity of PapS in a heterologous system, we produced PapS in *E. coli* and a straight mutant of the alphaproteobacterium *Caulobacter crescentus*. However, we did not observe any effect on cell shape. However, our work shows that the ability of PapS to induce cell curvature relies on the presence and proper localization of the Por39/41 complex. Therefore, a sophisticated expression system would be required to co-express *papS*, *por39* and *por41* concomitantly at high levels and in the correct ratio.

Since the establishment of such a system is not trivial, these experiments are beyond the scope of the present study.

Very minor issues

II. 264-65, should be PapS-mNG to match the figure.

Corrected.

In immunoblots in the supplement, arrowheads are frequently used but only described in the legend of Fig. S5.

We have now added a definition of the arrowheads to the legends of all panels showing immunoblots.

Fig. S11 uses asterisks for significance estimates that are not defined in the legend; the other figures show the p -value directly.

We have exchanged the asterisks for the actual p -values.

Reviewer #3

In this manuscript Pöhl et al. set out to describe a cell biological mechanism that the bacterium *Rhodospirillum rubrum* uses to generate its characteristic helical rod morphology. This manuscript has an impressive number of discoveries and uses a variety of experimental approaches including advanced fluorescence microscopy and structural biology approaches. In general, the authors should be congratulated for preparing an extensive manuscript clearly describing the role of (newly-named) PapS and interactions with Por41, Por39, and the elongasome. In addition to further details about their individual claims described below, there are two major areas that need to be resolved. In brief, (1) while the authors predict an ~15% rate of outer curve extension relative to inner curve extension, they do not see this in their HADA labeling. This could be because of a distinction between the *rate* of extension/expansion/growth as compared to the *amount* of growth. Indeed, if there is a difference in the local rate of expansion, one would expect shape *changes* as opposed to the observed shape *maintenance*. (2) the single particle tracking analysis was somewhat confusing and needs to be described further, as well as described more fully in the context of other elongasome tracking data. The role of an outer membrane complex to feed back on shape is incredibly fascinating, and represents a novel strategy that bacteria can utilize to pattern the geometric growth of their wall, thereby patterning their stereotypical shape to grow and thrive.

Given the length of this manuscript, this reviewer's comments have been organized into three major claims/discoveries made in this manuscript, and a follow-up section with other minor points and style suggestions. Except where noted, the description of the methods, approaches, and analysis are well described with sufficient detail.

Claim 1: There is an outer-membrane protein complex in *R. rubrum* that promotes cell curvature, and this complex includes the lipoprotein PapS and porins Por39 and Por41. The support for claim 1 is very strong, including genetic disruption of PapS resulting in straight cells and straight sacculi. These shapes can be rescued by expression of PapS from a plasmid. The authors do a great job explaining their interpretation that the conservation of PapS in other curved *Rhodospirillales* implies PapS function is not limited to *R. rubrum*.

Comment 1.1: Strengthening the claim to prove that PapS is used in other species would likely require genetic tools and culturability in those species, and is likely beyond the scope of this study.

Please see our response to a similar issue (point 4) raised by Reviewer #2.

Claim 2: Porins Por39 and Por41 and PapS form helical ribbon-like structures at the outer curve of the cell, with the localization of PapS mediated by the porins. The authors generate semi-functional fusions of PapS fused to few different fluorescent proteins, all of which restore some cell curvature. Modifications predicted to disrupt the membrane binding and peptidoglycan binding domains showed that peptidoglycan binding was required for mediating cell curvature and outer membrane binding was also somewhat important. Following up with these PapS domain disruptions, the authors also examined PapS localization in the presence of Por39/Por41 disruptions. Unfortunately, entirely deleting these porins was not possible, so the authors utilized point mutants to dissect the order of assembly and localization, concluding that Por41 has the dominant role in positioning PapS assemblies. Finally, the authors dissect the binding interfaces on Por39/Por41/PapS for the complex and are able to generate point mutants that disrupt the interface. Overall, the support for claim 2 is strong.

Comment 2.1: line 158 claims protein accumulation was less than in WT-fusion, however none of the Western blots show a loading control to substantiate any claims of the relative amount of protein in any lane. Ideally, these loading controls would be included in all blots, but in the absence of this, claims about the abundance of protein from one lane to another should be removed.

We have removed statements regarding differences in protein abundance from the text.

Comment 2.2: The authors claim these putative disruptions remove membrane binding and peptidoglycan binding but do not provide evidence that they do. For example, is the purified PapS able to bind purified sacculi but the PG binding deficient one is not?

The incorporation of a lipoprotein into the outer membrane strictly relies on the presence of a lipoprotein signal peptide and an N-terminal cysteine in the processed protein, which both were removed by exchanging the native signal peptide of PapS with the signal peptide of the soluble periplasmic protein DipM from *C. crescentus*. Therefore, it is safe to assume that this mutant variant of PapS no longer represents a lipoprotein. However, since PapS associates with the outer membrane not only directly through its lipid anchor but also indirectly through its highly stable interaction with Por39/41, there is no straightforward experimental approach to verify the loss of the direct membrane interaction for the lipid-anchor-free variant.

Following the reviewers' suggestion, we have now analyzed the peptidoglycan-binding activity of PapS. Previous work on OmpA from *Acinetobacter baumannii* has shown that OmpA-like domains associate with peptidoglycan by interacting with mDAP residues of uncrosslinked peptide side chains (Park *et al*, FASEB J, 2012). We have now purified the OmpA-like domain of PapS and verified its DAP-binding activity by microscale thermophoresis, whereas the R223A variant completely lacked DAP-binding activity (new **Figure 1d**). Interestingly, its affinity for DAP (K_d of 2.5 mM) is markedly lower than the one obtained for *A. baumannii* OmpA (K_d of 9 μ M), which we re-analyzed as a positive control. The dense packing of PapS in the vicinity of the peptidoglycan layer likely ensures that it still binds the cell wall with high efficiency. However, its looser association may be critical to facilitate the crosslinking of mDAP residues in regions occupied by PapS and the escape of elongasome complexes that are caged within PapS assemblies. A discussion of these points has now been added to the **Discussion** section.

Comment 2.3: There should be some discussion and/or acknowledgement that the fluorescent fusions do not restore cell curvature completely, and restore it to different amounts in different constructs. This is to be expected in fluorescent fusions, and should be discussed somewhere.

We now mention in the Results that the strain producing the PapS-mCherry in place of the native PapS protein shows a reduced sinuosity, which may be explained by steric hindrance by the fluorescent tag or the accumulation of degradation products. Please see also our response to a similar issue (point 2) raised by Reviewer #2.

Comment 2.4: There are a multiple prominent bands in S3B, S5B that run at different weights. There should be some, even minor, discussion of this because it implies to the reader that there are different complexes or protein abundances in these strains. In S3B, are these breakdown products? are those products fluorescent? How does this influence the interpretation of the fluorescent localization?

Please see our response to a similar issue (point 2) raised by Reviewer #2.

Minor Comment 2.5: The brief mention of FM4-64 is not as clearly defined as many of the other observations. Does this localization of FM4-64 depend on the Por39/Por41/PapS complex? In straight(er) cells that still have PapS with its membrane binding domain intact, is there preferential lipid stain? In straight cells lacking PapS entirely, does one face of the cell still have an FM4-64 enrichment?

We have now analyzed the localization pattern of FM4-64 in additional backgrounds (see new **Supplementary Figure 6d**). We found that the helical pattern is also observed in $\Delta papS$ cells and in cells carrying the D71S exchange in Por41, which results in porin delocalization. These findings indicate that the helical organization of the outer membrane is not mediated by the porin-PapS assemblies. It is tempting to speculate that, conversely, the porin-PapS assemblies could follow an intrinsic helical organization of outer membrane lipids and proteins in *R. rubrum*. However, the validity of this hypothesis remains to be determined.

Claim 3: These porin complexes trap the elongation machinery, leading to biased growth such that the outer curve grows faster than the inner curve, leading to overall cell curvature. The authors use a combination of tracking wall synthesis with fluorescent D-amino acids and single-particle tracking of various complexes to conclude that the elongosome components are trapped and spend more time on the outer curve. In addition to this HADA labeling, the authors went on to examine the localization and dynamics of a component of the elongosome, RodZ. This claim has the weakest support overall. In light of the conceptual issues described here, it is difficult to assess the author's overall model that porin complexes trap the elongosome.

Major comment 3.1: The authors comment that they expect a 15% increase in HADA staining on the outer curve, but do not observe any enrichment. This conclusion is complicated by the fact that *R. rubrum* maintains its relative curvature throughout the growth phase. While the outer curve may be 15% longer, it is not immediately obvious that the outer curve has to grow faster per unit surface area. This is a major point that needs to be resolved in the language used to describe the dynamics of shape establishment as compared to shape propagation. This also applies to the language and description in the overall working model of Figure 10. The authors do not demonstrate evidence of going from a straight rod to a helical rod, rather disrupted shapes that happily propagate with altered geometry. This same mechanism may be relevant for establishment of shape, but here the authors simply test shape maintenance. In regards to deciphering the role of PapS/Por39/Por41 in maintaining shape, the authors do an excellent job and should be commended on their logical progression and description of the system.

We agree that it is not necessary to increase the rate of peptidoglycan biosynthesis *per unit surface area* to maintain curved cell shape. We have now changed the wording in the Discussion and the Supplementary Results to make this clear.

Elongosomes move around the circumference of the cell body in helical tracks almost perpendicular to the long axis of the cell. We propose that their entrapment in the porin-PapS assemblies increases the time interval they spend moving in the outer curve during each helical turn they make, thereby leading to a relative (15%) increase in the total amount of peptidoglycan inserted *along the total length* of the outer curve of the cell.

Notably, this mechanism is sufficient both for the establishment and the maintenance of cell curvature. If a cell is straight, the density of elongosome tracks at two opposite sides is equal. In this case, an

increased incorporation of peptidoglycan at the porin-PapS assemblies results in a relative increase in the rate of cell elongation per unit surface area at this location and, thus, the establishment of cell curvature. As the cell body bends, the relative density of elongasome tracks at the outer curve becomes increasingly lower, while that at the inner curve remains constant. As a consequence, the difference in the rate of cell elongation per unit cell surface area between the outer and the inner curve keeps on decreasing, until it reaches a steady state in which both sides grow at the same rate, thus maintaining a constant degree of curvature. We have now incorporated this extended model in the Discussion.

Major comment 3.2: The single particle tracking analysis throughout was confusing. For many previously published studies of single particle tracking of elongasome components, trajectories were either measured in just the peripheral plane [<https://www.nature.com/articles/ncomms13170>, <https://elifesciences.org/articles/32471>] or in an unwrapped 3D surface of the cell [<https://www.pnas.org/doi/full/10.1073/pnas.1509610112>].

While it is true that our analytical approach differs from those used in previous work, this does not in any way invalidate our single-particle tracking (SPT) studies. The first two publications listed above investigate the dynamics of elongasome components in rod-shaped species in which elongasomes move freely around the circumference of the cell. However, the focus of the SPT analysis in our study is to understand the relationship between PapS and elongasome complexes (with RodZ as a proxy). As PapS localizes at the outer curve of each cell, it was not suitable for our purposes to measure dynamics exclusively at the peripheral plane. Instead, we decided to measure PapS dynamics at the center plane and RodZ dynamics at both the peripheral and the center plane. The third study mentioned above did not perform single-particle tracking but tracked the entire elongasome complex by analyzing Z-stacks that were recorded by widefield microscopy.

The authors need to include a more careful description of their analysis.

A comprehensive explanation of the experimental and analytical approaches used in our SPT analysis is given in the Materials section under the paragraph 'Single-particle tracking (SPT)'. Compared to other publications in the field, we have tried to be very explicit in the description of the single-particle imaging methods, and we are not sure what additional details Reviewer #3 would like us to include. The analytical approach used is not novel and relies on the widely used software SMTracker 2.0, which is fully described in two publications (<https://doi.org/10.1038/s41598-018-33842-9> and <https://doi.org/10.1093/nar/gkab696>). It would not be useful to describe the algorithms used by this software again in detail in our manuscript.

The bubble plots for mixed populations of dynamics appear to utilize a global definition of D1 and D2, and allow only a fitting parameter for the fraction of each population. However, the influence of cell shape, and sub-cellular compartmentalization on trajectories is not discussed in the current manuscript. See [<https://www.sciencedirect.com/science/article/abs/pii/S0378437106012039>, <https://elifesciences.org/articles/82654>, <https://www.pnas.org/doi/full/10.1073/pnas.1102255108>].

The comparison of protein dynamics across multiple strains/conditions was indeed performed using globally defined (simultaneous) apparent diffusion constants. The use of simultaneous constants is required when comparing subpopulation percentages across different conditions, as it allows for the direct detection of shifts in the sizes of distinct subpopulations. In this way, it was possible to determine the relative contributions of different intermolecular interactions to the overall mobility of PapS, porin and RodZ molecules. The global definition of diffusion constants in our study is now explicitly stated in the legend to **Figure 3f**. Moreover, we now refer to D1 and D2 as "apparent diffusion constants" to make clear that the values obtained may not reflect the individual diffusion constants obtained without the use of global definitions.

The influence of cell shape and subcellular compartmentalization on the trajectories measured was not taken into consideration. Accounting for said influence is not necessary, as we do not observe any correlation between changes in the diffusion behavior of proteins and changes in cell morphology. For instance, both an increase and a decrease in PapS-PAmCherry mobility can be observed in rod-shaped mutants in **Figure 6f**).

We would like to stress the fact that the analytical approach used here is comparative and not meant to determine exact diffusion coefficients, but rather to detect changes in the dynamic behaviour of the examined proteins.

The relative mobility of the elongasome complexes is confusing and should be more carefully compared to previous literature values. For example, is the slowly moving complex with a $D_{eff} = 0.0191 \mu\text{m}^2 \text{s}^{-1}$ truly moving? Or does this more closely match the faster two subpopulations observed in <https://elifesciences.org/articles/50629>?

An apparent diffusion coefficient of $0.0191 \mu\text{m}^2 \text{s}^{-1}$ (PapS) or $0.0174 \mu\text{m}^2 \text{s}^{-1}$ (RodZ – center plane) would be considered “immobile” when compared to the diffusion constants determined in the manuscript cited by Reviewer #3. We have therefore now replaced the term “fast-moving” with “diffusive” and the term “slow-moving” with “immobile”.

The apparent diffusion coefficients for mNG-RodZ determined in our work should not be directly compared to those reported for elongasome components in the abovementioned study, as they were measured at a slower frame rate and calculated based on a global definition of D1 and D2 that may not precisely reflect the diffusion coefficients obtained when performing the analysis independently for each condition. Furthermore, it is not possible to compare the mobility of the elongasome complex in the manuscript mentioned above (i.e., the directional movement of the elongasome driven by its cell wall biosynthetic activity) with that of mNG-RodZ in our work, because the time interval used to study mNG-RodZ dynamics (a few hundred milliseconds) is on a completely different scale than the one required to follow elongasome movement (up to one minute).

As a followup to needing more details about these procedures, the methods section describes the illumination for the tracking as being in “TIRF mode”, which would imply the illumination intensity as limited to the peripheral plane and not illuminate anything in the center plane.

While TIRF is generally used to limit illumination of the sample exclusively to the peripheral plane, the illumination depth can be modulated by changing the incident angle of the laser, as long as the incident angle remains larger than the critical angle. As a result, the illumination depth can usually range from 30 to 300 nm. In most of our experiments, we used TIRF mode at an angle (62°) that, in our hands and with our setup, allows the evanescent field to penetrate the sample to a depth of a few hundred nanometers, with the sole purpose of reducing the overall background. A larger angle (64°) was used exclusively to image mNG-RodZ in the peripheral plane (see **Figure 9**). It is important to note that the focal plane of the microscope also plays an important role in what is imaged. For instance, in the presence of an evanescent field that penetrates the sample to a depth of 300 nm, it is possible to record preferentially either peripheral or cytoplasmic signal simply by changing the focal plane accordingly. An intuitive visualization of how a change in the incident angle influences the penetration depth of the evanescent wave can be found at <https://micro.magnet.fsu.edu/primer/java/tirf/penetration>

In addition to comments and suggestions about these specific claims, here are a few minor stylistic changes that will improve the readability overall.

Minor comment 4.1: In addition to the sinuosity of delta PapS cells in Supp 1A, it would have been nice to see a delta PapS with an empty vector control in Supp 3A to ensure that plasmid bearing strains are straight unless they have PapS. It also appears as if the data for the WT cells is replotted/identical in Supp 1A and Supp 3A, which would be nice to include as a comment in the figure captions.

We have added an empty vector control to this analysis. The distribution of cell sinuosities measured for the control cells closely resembles the one obtained for $\Delta papS$ cells, which shows that the presence of *papS* on the plasmid is required to generate cell curvature.

In some of the supplemental figures, the sinuosity distributions of wild-type and $\Delta papS$ cells were replotted as a reference. We have now indicated this fact in the figure legends.

Minor comment 4.2: In supplemental figure 1E, recommend removing bar graph and using just the simple colored dots to represent each experiment. [<https://github.com/cxli233/FriendsDontLetFriends>]

We agree that bar plots alone are not suitable to represent the mean of a distribution. However, we additionally give the data points underlying the bar plots, so that the distributions are apparent. We would prefer to show the bars because they provide a better visualization of the sizes of the means.

Minor comment 4.3: line 55, Por41 and P39 => should be Por41 and Por39?

Corrected.

Minor comment 4.4: Figure S3CD, I was expecting representative phase images and quantification of the fluorescent strains as well.

We have now added phase contrast images and sinuosity distribution plots for representative strains producing PapS-mNG (SP02) and mCh-Por39 (SP222) in the presence and absence of cephalixin to **Supplementary Figure 3c,d**. For these strains there was no significant difference in cell sinuosity in the absence or presence of cephalixin.

Minor comment 4.5: Figure 3C, S5C, pValues are very confusing in the way they are presented, maybe add some heads to the lines, or make sure they line up appropriately? Also, if using scientific notation on some pValues, please use it on all.

We added small nubs to the lines to make it clearer which conditions are being compared. We chose to only display *p*-values smaller than 0.01 in scientific notation to visually distinguish samples that differ with high confidence from samples that are on the verge of being significantly different.

Minor comment 4.6: This reviewer experienced confusion in the characterization purified PapS and the differences between different PapS fusions. In general the authors do a good job clarifying the difference between fluorescent fusions and untagged proteins, but only include size exclusion chromatography for the non-fusion PapS construct.

The size exclusion experiment was performed to clarify the oligomeric state of native PapS. The results show that PapS only exists in the monomeric state under the tested conditions, which is in line with the observation that the PapS structures rapidly disintegrates after gentle cell lysis. The fluorescent proteins used to generate the different fusion constructs were specifically designed to not form dimers, which makes it highly unlikely that they affect the oligomerization state of PapS or the porins. If the fluorescent tags induced large-scale changes in the arrangement of the porin-PapS assemblies, they would be expected to severely affect the functionality of the system, which is not the case.

Minor comment 4.7: To help the reader contextualize the scale for sinuosity, maybe there could be a supplementary figure with a few toy examples of different sinuosity, or phase contrast images of individual cells as real examples that show a range from 1 to 1.05 or even 1.08.

Thank you for this suggestion. We have now added phase contrast images of representative cells with different sinuosities to **Supplementary Figure 1a**.

Minor comment 4.8: Please check for order of subfigures, it would be easier for the reader to have the subfigures remain in their current positions, and have labels that go in order, even if their reference in the main text does not. For example, Figure 4 has labels go a, b, c, d down the left, then jump up to e before returning next to d for f then g and h go left to right and end with i at the top. This is hard to follow. Also applies to other figures, such as Fig 9.

We are aware that the ordering of the panels is not optimal. However, journal policy usually requires figures to be labeled in the order in which they are referenced in the text. Since it would take up much more space to arrange the panels in the proper order, we would prefer to leave the figures as they are.

Minor comment 4.9: The label under the columns in S6AB was confusing. The combination of tilted names, and some underlined, and some not underlined was difficult to follow. Maybe the authors could try a different labeling strategy to clarify. This also applies to S5C, S7CDE, S11C although those ended up being less confusing for this reviewer to follow.

We have modified the labeling in **Supplementary Figure 6a,b** and hope that the new labeling strategy makes it easier to read the graph.

Minor comment 4.10: There is a small typo in the legend of figure 6. Subplot 'g' should be 'f' instead.

Corrected.

Reviewer #4 (Remarks to the Author):

Reviewer #2 (Remarks to the Author):

The revised version of this already excellent MS satisfies my previous (minor) concerns. Great work. Thank you.

Reviewer #3 (Remarks to the Author):

The updated manuscript addresses my concerns. I appreciate the authors expansion and edits in the writing of this draft as well as their thoughtful considerations and responses to the reviewers' requests. I continue to be excited about the discoveries described here and the way they advance our understanding of helical cell morphogenesis.

One remaining issue that the authors and this reviewer disagree on is the importance of cellular geometry to the detection and tracking of the SPT data. While I agree that altering the penetration depth can be achieved by changing the incident angle, the evanescent field will always be brightest at the TIR interface. Even though the apparent focal plane of the microscope is set to a different position (say 300 nm above the surface), it is still possible that event detection could still be biased by the intensity distribution of the evanescent field. This is not a major point, and I think the increased clarity in the methods section of this version provides readers with sufficiently context to follow the authors argument.

Minor point: Please ensure all scale bars have their length mentioned. Almost all do, but at least figure 2G is missing the length of the bar in the legend.

Reviewer #4 (Remarks to the Author):

Response to reviewers

We thank the three reviewers for taking the time to go through the revised version of our manuscript and evaluate the additions and changes we made. Please see below for our comments.

Reviewer #2

The revised version of this already excellent MS satisfies my previous (minor) concerns. Great work. Thank you.

Thank you, we appreciate your positive feedback.

Reviewer #3

The updated manuscript addresses my concerns. I appreciate the authors expansion and edits in the writing of this draft as well as their thoughtful considerations and responses to the reviewers' requests. I continue to be excited about the discoveries described here and the way they advance our understanding of helical cell morphogenesis.

Thank you for your positive feedback and your excitement about our work.

One remaining issue that the authors and this reviewer disagree on is the importance of cellular geometry to the detection and tracking of the SPT data. While I agree that altering the penetration depth can be achieved by changing the incident angle, the evanescent field will always be brightest at the TIR interface. Even though the apparent focal plane of the microscope is set to a different position (say 300 nm above the surface), it is still possible that event detection could still be biased by the intensity distribution of the evanescent field. This is not a major point, and I think the increased clarity in the methods section of this version provides readers with sufficiently context to follow the authors argument.

We agree that there could be a bias in the brightness of the particles observed at different distances from the TIR interface. However, when the focus is set to the center plane of the cells, the particles immediately adjacent to the TIR interface are out of focus and thus hardly detected. We obtained very different single-particle behaviors in the center and peripheral planes, suggesting that the particle populations observed in these two conditions are different, or at least show little overlap.

Minor point: Please ensure all scale bars have their length mentioned. Almost all do, but at least figure 2G is missing the length of the bar in the legend.

Thank you for making us aware of this issue. We checked all figures and added the missing information.

Reviewer #4

We appreciate the time and efforts you invested into re-reviewing this manuscript.